# LESS FORGETTING LEARNING: MEMORY-FREE CONTINUAL LEARNING CLASSIFICATION

## ABSTRACT

Continual Learning (CL) refers to a model's ability to sequentially acquire new knowledge across tasks while minimizing Catastrophic Forgetting of previously learned information. Many existing CL approaches face scalability challenges, often relying heavily on memory or a model buffer to maintain performance. To address this limitation, we propose "Less Forgetting Learning" (LFL), a memory-free CL framework for class and task incremental learning classification that does not rely on any **memory buffer**.

The LFL adopts a **stepwise freezing and fine-tuning** strategy. Different components of the network are trained in separate stages, with selective freezing applied to preserve critical knowledge. The framework leverages knowledge distillation to strike a balance between stability and plasticity during learning. Building upon this foundation, LFL+ incorporates an under-complete Auto-Encoder to preserve the most informative features. In addition, the LFL+ addresses the bias toward new classes in the classification head. Extensive experiments on three benchmark datasets show that LFL achieves competitive performance while requiring only 2.53% of the **model buffer** used by state-of-the-art methods. In addition, we propose a new metric designed to assess CL's plasticity-stability trade-off better.

## 1 INTRODUCTION

Neural Networks trained sequentially on multiple tasks often suffer from Catastrophic Forgetting, where new learning disrupts previously acquired knowledge (Wang et al., 2023). This challenge stems from the plasticity–stability dilemma. While Neural Networks exhibit high plasticity, enabling them to adapt to new tasks quickly, they often lack the stability needed to retain previously learned tasks. As a consequence, performance on earlier tasks tends to degrade over time (Vahedifar et al., 2025). Continual Learning (CL) provides a framework to mitigate Catastrophic Forgetting, allowing Neural Networks to learn sequentially. It is also known as lifelong, sequential, or incremental learning (Pfülb & Gepperth, 2019).

Various approaches have been proposed to solve Catastrophic Forgetting during CL, which can be broadly categorized as follows:

**1. Regularization-based**, methods introduce constraints to the loss function to prevent significant changes to the NN parameters crucial for previously learned tasks, such as: *Elastic Weight Consolidation (EWC)* (Kirkpatrick et al., 2017), *Synaptic Intelligence (SI)* (Zenke et al., 2017), *Learning without Forgetting (LwF)* (Li & Hoiem, 2018), and *Bias Correction (BiC)* (Wu et al., 2019).

**2. Memory-based** methods rely on mechanisms that encode, store, and retrieve past information to mitigate forgetting such as: *Incremental Classifier and Representation Learning (iCaRL)* (Rebuffi et al., 2017), *Dark Experience Replay++ (DER++)* (Buzzega et al., 2020), *PODNet* (Douillard et al., 2020), *CO-transport for class Incremental Learning (Coil)* (Zhou et al., 2021), and *Gradient Episodic Memory (GEM)* (Lopez-Paz & Ranzato, 2017).

**3. Dynamic Architecture** methods adaptively modify the model's structure to accommodate new tasks. This is typically achieved by expanding the network—e.g., adding new layers or modules—while keeping earlier components fixed to prevent interference such as: *Progressive Neural Networks (PNN)* (Rusu et al., 2022), *DyTox* (Douillard et al., 2022), and *Memory-efficient Expandable MOdel (MEMO)* (Zhou et al., 2023).

For an algorithm to be considered a CL method, it must adhere to several key criteria:

**1.** Balanced Learning: The model should maintain performance across both previously learned and newly introduced classes (tasks), preventing overfitting to recent classes (tasks) (Wu et al., 2019).

**2.** Learning from Stream of Data Sets: The algorithm should be capable of learning from a stream of data sets with more recent data sets introducing new classes (Rebuffi et al., 2017).

**3.** Scalability and Efficiency: Computational and model buffer overhead should remain stable or grow minimally with the number of tasks.

Despite significant advances in CL, many existing CL methods do not fully comply with these criteria, such as (Kirkpatrick et al., 2017; Zeng et al., 2019; Saha et al., 2021). They rely on *memory buffers* to store and retrieve data from previous tasks, which represents an apparent weakness as it violates the strict CL setting, where no prior task data should be accessible (Buzzega et al., 2020). The fundamental limitation of relying on past data is that their memory requirements grow as new tasks are introduced. The critical question is: How can we design a memory-efficient CL framework that operates within the fixed capacity of the backbone network without sacrificing performance?

This paper presents a novel CL approach, *Less Forgetting Learning (LFL)*, along with its enhanced variant, the *LFL+*. Both methods extend the *Knowledge Distillation (KD)* framework while maintaining a memory-free learning paradigm. Memory-free means they do not rely on any external memory buffer (i.e., data from previous tasks); nevertheless, they employ a model buffer. The LFL framework decomposes the NN into three key sub-components:

**1. Shared parameters**, which serve as the main feature extraction backbone.

**2. Old task heads**, which represent previously learned classifiers.

**3. New task head**, a newly added classifier for the current task.

The central motivation can be understood through an analogy: the old and new task heads are like two students who must work together as a team, while the shared parameters act as their teacher. Both students learn by drawing knowledge from the same teacher, but the introduction of a new student (the new task head) should not significantly disrupt the progress of the existing student (the old task head). To support this balance, we employ a stepwise freezing strategy, which anchors the learning of both task heads at different stages and helps them collaborate effectively. Importantly, the teacher, when instructing the new student, does not have direct access to the old student's learning data; instead, knowledge transfer must occur indirectly through the teacher.

The core contribution of our work is a simple approach to mitigate Catastrophic Forgetting without modifying the network architecture, introducing dynamic components, or storing samples from previous tasks. We identify a potential to overcome this challenge solely by leveraging the over-parameterized nature of Neural Networks (Frankle et al., 2020) and introduce a stepwise LFL procedure. The intuition is that guiding different parts of the network to learn at different stages enables the model to strategically leverage a network's overparameterization (Fig. 1). This process sequentially trains and freezes specific components of a multi-headed network (consisting of a shared teacher and task-specific student heads) to isolate knowledge acquisition and prevent interference.

The procedure unfolds in four distinct phases: **1.** Initial Task Training: A shared teacher network and a corresponding student head for the first task are trained on the initial dataset. **2.** New Task Introduction: Upon the arrival of a new task, a new student head is added. The teacher network and the old student head are frozen. This prevents the new task from overwriting previously acquired knowledge in the shared backbone. **3.** Teacher & Old Head Alignment: The new student head has learned its new task. Concurrently, the teacher and the old student head are unfrozen to adapt to the new student, but the new student's head is now frozen. This prevents changes to the shared parameters from negatively impacting the newly learned knowledge. **4.** Final Consolidation: The teacher and both student heads are unfrozen to consolidate knowledge. The teacher provides a consistent representation, while the old student head is encouraged to retain the best version of the knowledge it has learned for its respective task concurrently with its adapted head.

The LFL+ further incorporates an additional step on the LFL by an Auto-Encoder for feature retention. We introduce Auto-Encoder as a mechanism for preserving knowledge from previously learned tasks while adapting to new ones. For each task, an under-complete Auto-Encoder is trained

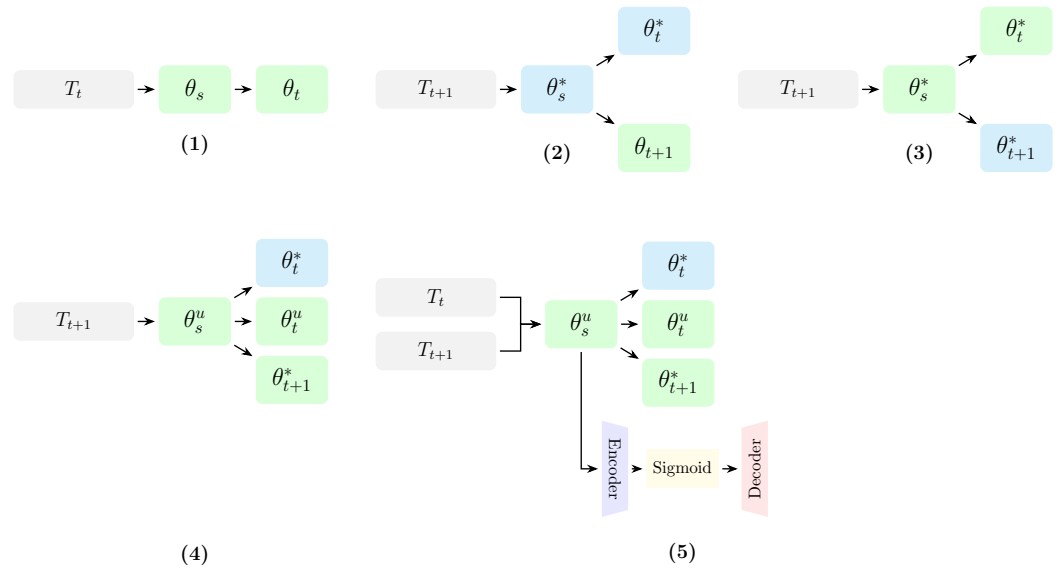

Figure 1: The stepwise freezing and fine-tuning pipeline of the LFL (4-steps) and LFL+ (5-steps). Each step's output serves as the next step's input. Gray boxes indicate task-specific data; Green boxes represent parameters trained in the current step; Blue boxes denote frozen (non-updated) parameters. **(Step 1)** Initialize training on task $T_t$ with $(\theta_s, \theta_t)$ yields trained parameters $(\theta_s^*, \theta_t^*)$. **(Step 2)** A new task $T_{t+1}$ is learned by freezing the previously trained parameters $(\theta_s^*, \theta_t^*)$ and training a new task-specific head $\theta_{t+1}$. **(Step 3)** Knowledge distillation is applied to update $(\theta_s^*, \theta_t^*)$, while $\theta_{t+1}^*$ is frozen. Here, logits $H_t$ are soft targets. **(Step 4)** Fine-tuning is performed jointly with final refinement that uses two soft targets, original logits $H_t$ and updated logits $H_t'$. **(Step 5)** LFL+ integrates an Auto-Encoder trained on shared representations from $T_t$ to preserve the most informative features. The Auto-Encoder helps regularize adaptation to $T_{t+1}$ and provides a transformation to correct bias caused by class imbalance in the last layer.

after the corresponding task model. This Auto-Encoder captures the most relevant features required for that task's objective. When a new task is introduced, the Auto-Encoder ensures that these critical features are preserved by enforcing a reconstruction loss. In doing so, only a subset of features is constrained to remain unchanged. At the same time, the rest of the model retains the flexibility to adapt to the new task using its remaining capacity. In addition, the LFL+ addresses the bias in the final classification layer, which tends to new classes as they have accessible samples. By applying a bias mitigation technique, the LFL+ promotes more balanced predictions and ensures stable performance across evolving task distributions.

## 2 RELATED WORK

Numerous methods leverage knowledge distillation (Hinton et al., 2015) to mitigate Catastrophic Forgetting by designating a previous version of the model as a *teacher*, and the current model as a *student*. Student model learns from targets provided by the teacher model, thereby capturing nuanced patterns that enhance the student model's generalization ability (Gou et al., 2021). KD-based methods that rely on memory buffers raise concerns regarding data privacy, memory efficiency, and computational overhead. These constraints make them impractical for applications with strict storage limitations or regulations prohibiting exemplar storage.

Among early approaches, LwF (Li & Hoiem, 2018) utilizes a shared convolutional network across tasks, modifying cross-entropy loss to retain prior predictions. While effective, it struggles with performance drops when new tasks come from different distributions. iCaRL (Rebuffi et al., 2017) demonstrates that LwF suffers from error accumulation in sequential learning scenarios where data originates from the same distribution. iCaRL attempts to mitigate these issues by storing a subset of

Table 1: Notation for the LFL and the LFL+ methods

| Symbol | Description | Symbol | Description | Symbol | Description |
|--------|-------------|--------|-------------|--------|-------------|
| | $A \in \{X, Y, O, H, \theta\}$ | $\theta$ | NN parameters | $i$ | Step number |
| $A_C^B$ | $B \in \{i, u\}$ | $u$ | **u**pdated parameters | $t$ | previous task |
| | $C \in \{t, t+1, s\}$ | $s$ | **s**hared parameters | $t+1$ | current task |
| $X$ | Data samples | $Y$ | Data samples hard targets | $O$ | NN **O**utput targets |
| $H$ | Soft Targets | | | | |

data from previous tasks. Similarly, DER++ (Buzzega et al., 2020) enhances knowledge retention by combining KD with cross-entropy loss to maintain inter-class relationships. Coil (Zhou et al., 2021) proposes implementing bidirectional distillation through co-transport, leveraging the semantic relationships between the previous and updated models to enhance knowledge retention and transfer. In addition, dynamic network-based methods such as PNN (Rusu et al., 2022), DyTox (Douillard et al., 2022), and MEMO (Zhou et al., 2023) tackle memory constraints in CL by expanding models efficiently. They observe that shallow layers across tasks remain similar, whereas deeper layers require greater adaptation, optimizing network expansion with minimal resource overhead.

## 3 LESS FORGETTING LEARNING

We advocate that CL methods should ideally avoid accessing data from previous tasks entirely. The LFL introduces a straightforward optimization process for training by leveraging the NN parameters trained on previous tasks. In the LFL+, we incorporate an Auto-Encoder into the LFL. Table 1 summarizes the notation that we used for the LFL and the LFL+.

### 3.1 DESIGN PHILOSOPHY

Our approach is fundamentally motivated by the Lottery Ticket Hypothesis (Frankle et al., 2020), which demonstrates that Neural Networks are significantly overparameterized. While standard approaches, such as weight pruning (Frankle et al., 2020) and neuron pruning (Ghorbani & Zou, 2020), often seek to compress these networks to 10–20% of their original size, we investigate the inverse property: can this inherent redundancy be repurposed to support CL? Instead of introducing external memory buffers or expanding the architecture, we aim to maximize the utility of the existing capacity. We hypothesize that by carefully orchestrating which parameters are active at each stage, we can mitigate Catastrophic Forgetting by ensuring that the network's "spare" capacity is utilized to stabilize training rather than being pruned or overwritten.

**Strategy: Stepwise Freezing via Component Decomposition:** To operationalize this, we decompose the Neural Network into three logical components: the shared backbone ($\theta_s$), the old task head ($\theta_t$), and the new task head ($\theta_{t+1}$). By treating each component as a binary state—Train ($T$) or Frozen ($F$)—we identify a search space of $2^3 = 8$ possible training configurations. We analyze these combinations to select a sequence that minimizes interference$\{\theta_s, \theta_t, \theta_{t+1}\}$:

*1. FFF:* The static state; no learning occurs.
*2. TFF:* Updates the backbone without allowing classifiers to adapt. This creates a misalignment between features and decision boundaries, rendering the update useless.
*3. FTF:* Fine-tunes the old head on fixed features. This yields no improvement for new task.
*4. FTT:* Updates only the heads. With a frozen backbone, the model lacks the plasticity to learn representations for the new task, resulting in negligible performance gains.
*5. TFT:* Updates the backbone and new head while freezing the old head. Here, gradients from new head ($\theta_{t+1}$) propagate into the backbone ($\theta_s$). This aggressively alters features required by the frozen old head ($\theta_t$), directly causing catastrophic forgetting.

Based on this analysis, we discard the suboptimal configurations and arrange the remaining three effective configurations (*6. FFT*, *7. TTF*, and *8. TTT*) into a sequence that preserves previous acquired knowledge:

**Step 2: Isolation of the New Task Head** (*6. FFT*): This step is essential for preventing immediate destabilization of learned features. As noted in the *TFT* analysis, joint training would propagate

noisy gradients into the backbone. We freeze the backbone and old head $(\theta_s^*, \theta_t^*)$ and train only $\theta_{t+1}$. This relies solely on the existing feature representations to "initialize" the new capability without corrupting the old one.

**Step 3: Controlled Feature Adaptation** (*7. TTF*): This step is essential for adapting the backbone for the new task while anchoring it to the old task's logic. The backbone must eventually update to accommodate the complexity of both tasks. We freeze the new task head $(\theta_{t+1}^*)$ and update $\theta_s^*$ and $\theta_t^*$. By freezing the new head, we remove the source of aggressive gradients. The update is dominated by the distillation loss from the old task, ensuring feature extraction adapts only in directions that do not violate the decision boundaries of the previous task.

**Step 4: Joint Fine-Tuning** (*8. TTT*): Fully joint training (equivalent to standard LwF). While necessary for final integration, applying this immediately allows the new task to dominate the shared representation. However, this step is essential for a global update to align the shared representation after Steps 2 and 3 optimize components locally. We unfreeze all components $(\theta_s^u, \theta_t^u, \theta_{t+1}^*)$. To prevent the forgetting associated with standard *TTT*, we employ two soft targets (from Step 1 and Step 3).

### 3.2 LESS FORGETTING LEARNING

Let us decompose our NN into two parts. Each NN can be seen as $\text{NN}^i(\theta_s, \theta_t)$. Therefore, $\theta_s$ denotes all shared parameters (feature extraction), and $\theta_t$ denotes the parameters of the last layer (classifier or a segmentation operator). Note, trained $\text{NN}^i$ target outputs can be denoted by $O^i$.

In **step one**, the $\text{NN}^1$ has been trained on a dataset $D_t(X_t, Y_t)$ describing a task $T_t$ formed by $c_t$ classes in the class set $\mathcal{C}_t$. $X_t$ denotes the data samples, each followed by a class label belonging to one of the $c_t$ classes, and $Y_t$ is the matrix formed by the one-hot class vectors corresponding to the targets for training the $\text{NN}^1$. The outputs of the $\text{NN}^1$ are obtained by following, loss function:

$$\mathcal{L}^1 = \mathbf{E}\Big[\mathcal{L}_{\text{CE}}(Y_t, O_t^1)\Big]. \tag{1}$$

Here, CE stands for cross-entropy loss function. The subscript in the output (i.e., $O$) denotes which set of classes is being learned, and the superscript indicates step numbering. After training the $\text{NN}^1$ on the dataset $D_t(X_t, Y_t)$, a new dataset $D_{t+1}(X_{t+1}, Y_{t+1})$ describing a new task $T_{t+1}$ is obtained and we aim to train a new NN model that can achieve high performance on the combined task $T = T_t \bigcup T_{t+1}$ formed by $c$ classes in the class set $\mathcal{C} = \mathcal{C}_t \bigcup \mathcal{C}_{t+1}$, without access to the $D_t(X_t, Y_t)$. LFL follows a four-step process, summarized in Algorithm 1 in the Appendix. We introduce the data samples $X_{t+1}$ to the trained $\text{NN}^1$ to obtain the logits (i.e., the outputs of the $\text{NN}^1$ before the softmax activation of the last layer):

$$\text{NN}^1(X_{t+1}, \theta_s^*, \theta_t^*) \xrightarrow{\;Computing\;} H_t. \tag{2}$$

These outputs will be used as *soft targets* for the new data $X_{t+1}$, providing knowledge concerning the old task $T_t$ and preserving parameters of the $\text{NN}^1$ leading to high performance on $T_t$.

For **step two**, we train the $\text{NN}^2$ only for new task head $\theta_{t+1}$ until convergence, where we freeze $\theta_s^*$ and $\theta_t^*$ obtained in step one (See part (2) of Fig. 1). The outputs of step two are obtained by following, loss function:

$$\mathcal{L}^2 = \mathbf{E}\Big[\mathcal{L}_{\text{CE}}(Y_{t+1}, O_{t+1}^2)\Big]. \tag{3}$$

Here, $Y_{t+1}$ denotes ground true labels for task $T_{t+1}$ classes.

For **step three**, we train the $\text{NN}^3$ model by training the feature extractor and the previously learned task head while freezing the output parameters $\theta_{t+1}^*$ obtained from step two (See part (3) of Fig. 1). The output of the old task head and the new task head will be $O_t^3$ and $O_{t+1}^3$, respectively. For this step, we calculate the following loss function:

$$\mathcal{L}^3 = \mathbf{E}\Big[\underbrace{\mathcal{L}_{\text{KD}}(H_t, O_t^3)}_{\text{Soft Targets}} + \underbrace{\mathcal{L}_{\text{CE}}(Y_{t+1}, O_{t+1}^3)}_{\text{Hard Targets}}\Big], \tag{4}$$

where KD stands for the Knowledge Distillation loss function defined as follows:

$$\mathcal{L}_{\text{KD}}(H_t, O_t) = -\sum_i h_i \log o_i, (a), \qquad h_i = \frac{H_i^{1/p}}{\sum_j H_j^{1/p}}, (b), \qquad o_i = \frac{O_i^{1/p}}{\sum_j O_j^{1/p}}, (c). \tag{5}$$

Here, $p$ represents a temperature parameter used to control the smoothness of the probability distribution, with values typically set to $p > 1$ (Hinton et al., 2015).

For **step four**, we aim to balance stability and plasticity. First, we compute updated references $H'_t$ by passing the data $X_{t+1}$ through the feature extractor updated ($\theta^u_s$) in Step 3, combined with the original old-task classifier ($\theta_t$) from Step 1:

$$\text{NN}^4\big(X_{t+1}, \theta^u_s, \theta_t\big) \xrightarrow{Computing} H'_t. \tag{6}$$

Here $u$ in the superscript of the parameters (i.e., $\theta$) indicates an **u**pdated version of step three parameters. For the last step, we train the $\text{NN}^4$ model, on the new dataset to learn from the new data with two logit targets (See part (4) of Fig. 1). We then train the full model $\text{NN}^4$ (fine-tuning all parameters). The loss function combines two distillation terms one preserving the original model's response ($H_t$) and one preserving the Step 3 model's response ($H'_t$) with the new task supervision:

$$\mathcal{L}^4 = \mathbf{E}\Big[\underbrace{\alpha\mathcal{L}_{\text{KD}}\big(H_t, O^4_t\big)}_{\text{Original Stability}} + \underbrace{(1-\alpha)\mathcal{L}_{\text{KD}}\big(H'_t, O'^4_t\big)}_{\text{Recent Stability}} + \underbrace{\mathcal{L}_{\text{CE}}\big(Y_{t+1}, O^4_{t+1}\big)}_{\text{New Task Plasticity}}\Big], \tag{7}$$

where $\alpha$ is hyperparameter. Note, the outputs of the old task head with old logit, the old task head with new logit, and the new task head will be $O^4_t, O'^4_t$, and $O^4_{t+1}$, respectively.

### 3.3 LESS FORGETTING LEARNING+

LFL + follows the same four-step process in LFL except for adding the Auto-Encoder before training $T_{t+1}$, summarized in Algorithm 2 in the Appendix and shown in part (5) of Fig. 1. The key idea is to preserve the most informative features by learning from the extracted features of $T_t$. At the start of training $T_{t+1}$, the model's feature extractor, $\theta_s$, has already been optimized for $T_t$. Subsequently, retaining only the most relevant features to the previous tasks while allowing flexibility for the rest to change improves the model's ability to adapt to the new task. We utilize an Auto-Encoder trained on the data representations from the previous task to achieve this. The Auto-Encoder function is formally defined as:

$$R(P) = W_{\text{Dec}}\sigma\big(W_{\text{Enc}}P\big), \qquad P = \theta_s(X_t). \tag{8}$$

Here $W_{\text{Enc}}$ is encoder weights, $W_{\text{Dec}}$ is decoder weights, $\sigma$ is the activation function. By regulating the distance between the reconstructed representations, the features retain the flexibility needed to adapt to variations introduced by the $T_{t+1}$ while preserving critical information for the previous task. So after **step one** and training task $T_t$ and obtaining the shared parameters $\theta_s$ and task-specific parameters $\theta_t$, in **step two** we train an under-complete Auto-Encoder with the following minimization objective loss function problem:

$$\arg\min_R \mathbf{E}_{(X_t, Y_t)}\Big[\Omega\big\|R(P) - P\big\|^2_2 + \mathcal{L}_{\text{CE}}\big(\theta_t\big(R(P)\big), Y_t\big)\Big], \tag{9}$$

where $\Omega$ is a hyperparameter. In the subsequent steps, we adopt the same approach utilized in the LFL framework (i.e the LFL+'s **steps 3 and 4** correspond to the LFL's **steps 2 and 3**, respectively).

In **Step 5**, we address the prediction bias towards new classes caused by data imbalance between the old and new classes. We introduce a logit transformation $\Gamma$ dependent on the encoder's response:

$$\Gamma(P_{t+1}) = w_{\text{bias}}\sigma(W_{\text{Enc}}P_{t+1}) + b_{\text{bias}} \quad (a), \quad P_{t+1} = \theta_s(X_{t+1}) \quad (b), \quad \widetilde{H}_t = \Gamma(P_{t+1}) \odot H_t \quad (c). \tag{10}$$

Here, $H_t$ are the stored logits from the old model and $\odot$ denotes point-wise multiplication. Note, $w_{\text{bias}}$ and $b_{\text{bias}}$ are in the same dimension with $W_{\text{Enc}}F_{t+1}$. These parameters are trained using a cross-validation set. Notably, all other parameters of the model remain frozen during this process. The minimization function for training the bias correction layer is in the following:

$$\mathbf{E}_{\text{bias}}\Big[\big\|\Gamma(X_{t+1}) - \mathbf{1}\big\|^2_2\Big]. \tag{11}$$

This promotes unbiased learning by guiding the transformation function to remain close to the identity mapping. The minimization objective function for training the Auto-Encoder is:

$$\mathbf{E}_{\text{Auto-Encoder}}\Big[\big\|\sigma\big(W_{\text{Enc}}\theta^r_s(X_{t+1})\big) - \sigma\big(W_{\text{Enc}}\theta_s(X_{t+1})\big)\big\|^2_2\Big]. \tag{12}$$

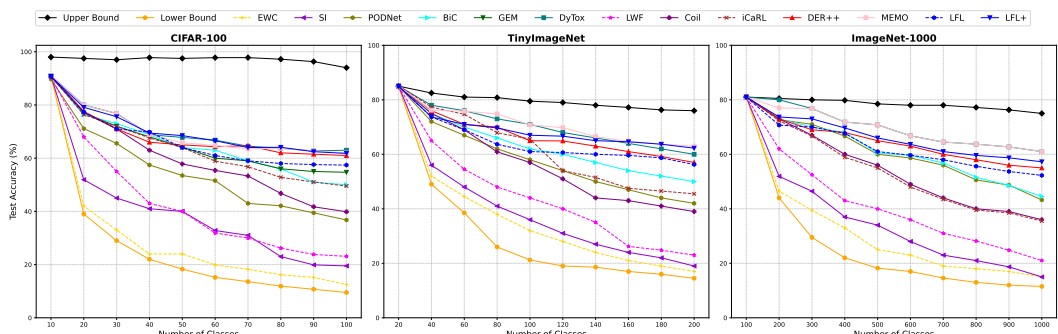

Figure 2: ACC evaluation comparison for the CIL for 10 tasks for each dataset.

The final loss function for training the LFL+, **step five** is:

$$\mathcal{L}^5 = \mathbf{E}\Bigg[ \underbrace{\eta \mathcal{L}_{\text{KD}}\Big(\widetilde{H}_t, O_t^5\Big)}_{\text{Bias-Corrected Stability}} + \underbrace{(1 - \eta)\mathcal{L}_{\text{KD}}\Big(H'_t, O'^5_t\Big)}_{\text{Step 3 Model Stability}} + \underbrace{\mathcal{L}_{\text{CE}}\Big(Y_{t+1}, O^5_{t+1}\Big)}_{\text{New Task Plasticity}}$$
$$+ \underbrace{\left\|\sigma\big(W_{\text{Enc}}\theta_s^r(X_{t+1})\big) - \sigma\big(W_{\text{Enc}}\theta_s(X_{t+1})\big)\right\|_2^2}_{\text{Feature Space Constraint}} + \underbrace{\left\|\Gamma(X_{t+1}) - \mathbf{1}\right\|_2^2}_{\text{Bias Regularization}}\Bigg]. \tag{13}$$

where $\eta$ is hyperparameter. Note, the outputs for the old task head with bias corrected logit, old task head with new logit, and the new task head will be $O_t^5, O'^5_t$, and $O^5_{t+1}$, respectively.

## 4 EXPERIMENT & DISCUSSION

**Scenarios:** We conducted a series of experiments to evaluate the performance in Task Incremental Learning (TIL) and Class Incremental Learning (CIL) scenarios (See details in Appendix). Physically, our model utilizes a single output layer matrix $W_{\text{out}} \in \mathbb{R}^{(C_{\text{old}} + C_{\text{new}}) \times P}$, where P is the feature dimension. However, logically, we treat this matrix as a concatenation of two blocks: $\theta_t$ (columns corresponding to previously learned classes) and $\theta_{t+1}$ (columns corresponding to new classes).

The breakdown of training into steps relies on masking specific gradients rather than separating architectures. In last step the entire single head is fine-tuned jointly, but the loss function remains logically split: $\mathcal{L}_{\text{KD}}$ is applied to the logits indexed by the old task, and $\mathcal{L}_{\text{CE}}$ is applied to the logits indexed by the new task. Consequently, task identifiers are strictly a *training-time* requirement used to define the boundaries of these logical partitions for loss computation. During inference, particularly in the Class-IL setting, the partitions are ignored, and the single head functions as a global classifier over the union of all classes $\mathcal{C}_t \cup \mathcal{C}_{t+1}$.

**Evaluation Metrics:** To assess the ability of each method to perform effective CL and battle Catastrophic Forgetting, we use Average Accuracy (ACC), Forward transfer (FWT), Backward transfer (BWT) for the TIL scenario and ACC, Average Forgetting (AF), and Intransigence (I) for the CIL scenario. We define each evaluation metric in the Appendix.

In addition, we propose a new metric: the Plasticity-Stability (PS) ratio. The main idea is that effective CL requires a balance between plasticity, which allows the system to acquire new knowledge, and stability, which ensures that previously learned knowledge is retained. Our proposed metric assesses how well CL methods scale with the increasing number of tasks and classes.

When CL methods trained for task $T_k$ on $D_k$, its accuracy on all tasks is measured using the corresponding test sets, leading to a matrix $A \in \mathbb{R}^{T \times T}$ containing the accuracies on all $T$ tasks, i.e., $A_{i,j}$ denotes the accuracy of the model on task $T_j$ after trained completely on task $T_i$.

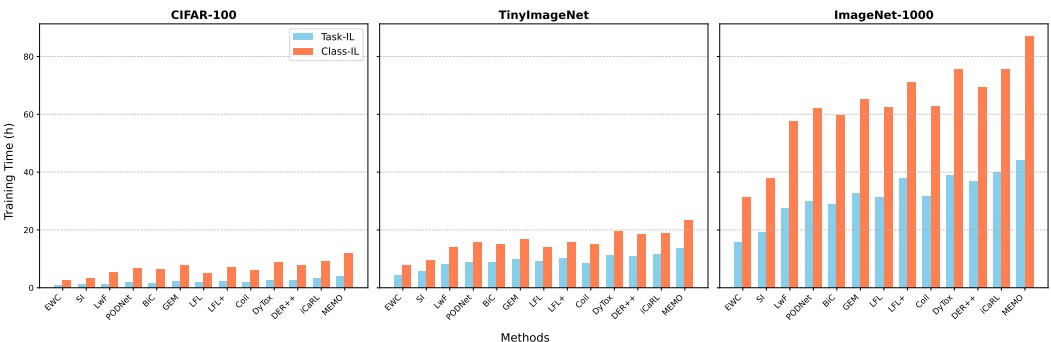

Figure 3: Training time (h) for CIL and TIL scenario. For buffer-based methods, a total of 5120 exemplars are utilized in TIL experiments, while CIL experiments employ 2000 exemplars for CIFAR-100 and Tiny-ImageNet, and 20,000 exemplars for ImageNet-1000.

**Plasticity-Stability (PS)**: This metric quantifies the trade-off between plasticity and stability:

$$\text{PS}_T = \frac{1}{T-1} \frac{\sum_{k=2}^{T} \left( A_{k,k} - A_{k-1,k} \right)}{\left| \sum_{k=1}^{T-1} A_{T,k} - A_{k,k} \right|}. \tag{14}$$

**Methods:** We compare the LFL and the LFL+ performance with other CL methods, and their characteristics are summarized in Table 4. We used Stochastic Gradient Descent (SGD) as Lower Bound (LB) and joint training as Upper Bound (UB) to establish performance bounds.

**Datasets:** We conduct experiments on CIFAR-100 (Krizhevsky, 2009) (100 classes, 10 tasks, 10 classes/task), Tiny-ImageNet (Le & Yang, 2015) (200 classes, 10 tasks, 20 classes/task for CIL), and ImageNet-1000 (Krizhevsky et al., 2012) (1,000 classes, 10 tasks, 100 classes/task for CIL)(See details in Appendix for 20 task for each dataset).

**Experiment Setup:** For experiments on the CIFAR-100, Tiny-ImageNet, and ImageNet-1000 datasets, we adopt the ResNet-18 architecture (He et al., 2016). All network layers are initialized using He initialization (He et al., 2015), which is well-suited for ReLU activations and ensures stable gradient propagation. To maintain consistency across CL methods, all models are trained using the SGD optimizer, while the Auto-Encoder is trained separately using the Adam optimizer (Kingma & Ba, 2015). The Auto-Encoder has a compact size of 1.56 MB (3.5%) compared to the 44.68 MB size of the main network. We perform an extensive grid search to fine-tune all hyperparameters. Table 11 presents the full set of evaluated hyperparameter configurations, while Table 12 and Table 13 report the best settings identified through this search. Additionally, 10% of the training data from each task is reserved as a validation set, which is used for bias correction and hyperparameter selection. Training is performed for a maximum of 100 epochs on CIFAR-100 and Tiny-ImageNet, and 200 epochs on ImageNet-1000, with early stopping based on validation loss. The batch size is fixed to 64 for all experiments.

To ensure robustness, all results are averaged over ten different random seeds. For memory-based methods in the CIL setting, we use 1,000 and 2,000 exemplars for CIFAR-100 and Tiny-ImageNet, and 10,000 and 20,000 exemplars for ImageNet-1000 (Zhou et al., 2024b). In the TIL setting, we follow prior work (Buzzega et al., 2020) and set 200, 500, and 5120 exemplars for all datasets. All experiments are conducted on a single NVIDIA A6000 GPU.

### 4.1 Comparison based on the CIL Scenario

As shown in Fig. 2, LFL+ achieves competitive or superior performance in the CIL scenario, particularly when compared to MEMO and DyTox. Although dynamic architecture-based methods outperform LFL and LFL+, and memory-based methods achieve comparable accuracy, both rely on significantly larger model or memory footprints. This raises significant concerns regarding computational scalability and practical deployment in resource-constrained environments with large-scale

Table 2: Results on CIFAR-100 for CIL scenario with 5 and 10 incremental classes.

| Buffer | Method | 5 Classes | | | | 10 Classes | | | |
|---|---|---|---|---|---|---|---|---|---|
| | | ACC↑ | AF↓ | I↓ | PS↑ | ACC↑ | AF↓ | I↓ | PS↑ |
| No Buffer | LFL+ | 66.97 | 0.1708 | -0.0054 | 0.5252 | 67.18 | 0.1289 | -0.0089 | 0.4287 |
| | LFL | 62.07 | 0.4103 | 0.0002 | 0.4551 | 63.21 | 0.3412 | -0.0081 | 0.3654 |
| | LwF | 31.23 | 0.5978 | 0.0048 | 0.2105 | 40.82 | 0.4817 | -0.0261 | 0.3641 |
| | SI | 31.79 | 0.5790 | 0.0014 | 0.2591 | 37.15 | 0.4159 | 0.0203 | 0.2801 |
| | EWC | 17.97 | 0.7287 | -0.0049 | 0.2362 | 27.41 | 0.7562 | -0.1095 | 0.2554 |
| 1000 | DyTox | 62.61 | 0.1579 | -0.0133 | 0.3790 | 67.19 | 0.1542 | -0.0022 | 0.4097 |
| | MEMO | 62.33 | 0.3074 | -0.0119 | 0.4407 | 66.39 | 0.2287 | -0.0019 | 0.2094 |
| | DER++ | 60.06 | 0.1969 | 0.0254 | 0.2992 | 64.82 | 0.1219 | 0.0115 | 0.2489 |
| | GEM | 59.98 | 0.1971 | 0.0258 | 0.3711 | 62.18 | 0.1238 | 0.0112 | 0.4012 |
| | BiC | 57.94 | 0.1871 | 0.1502 | 0.3652 | 61.83 | 0.1521 | 0.4582 | 0.3948 |
| | iCaRL | 58.98 | 0.3582 | -0.0193 | 0.2489 | 60.74 | 0.2471 | -0.0027 | 0.3052 |
| | Coil | 53.84 | 0.4008 | -0.0459 | 0.2562 | 56.42 | 0.3165 | -0.0128 | 0.3681 |
| | PODNet | 44.68 | 0.4173 | 0.1087 | 0.3336 | 51.84 | 0.3721 | 0.1032 | 0.3607 |
| 2000 | DyTox | 67.08 | 0.1476 | -0.0147 | 0.4006 | 70.52 | 0.1377 | -0.0025 | 0.4331 |
| | MEMO | 66.97 | 0.2876 | -0.0166 | 0.4731 | 69.92 | 0.2043 | -0.0029 | 0.2241 |
| | DER++ | 64.35 | 0.1841 | 0.0237 | 0.3215 | 68.41 | 0.1091 | 0.0129 | 0.2667 |
| | GEM | 64.15 | 0.1843 | 0.0241 | 0.3912 | 65.66 | 0.1099 | 0.0126 | 0.4229 |
| | BiC | 62.08 | 0.1749 | 0.1403 | 0.4038 | 65.19 | 0.1375 | 0.4977 | 0.4365 |
| | iCaRL | 63.19 | 0.3348 | -0.0212 | 0.2673 | 63.97 | 0.2243 | -0.0031 | 0.3263 |
| | Coil | 57.67 | 0.3748 | -0.0505 | 0.2752 | 59.67 | 0.2880 | -0.0143 | 0.3926 |
| | PODNet | 47.88 | 0.3900 | 0.1015 | 0.3607 | 55.02 | 0.3418 | 0.1156 | 0.3899 |

datasets or growing task granularity. Further analysis of KD-based methods, such as Coil, BiC, iCaRL, and DER++, reveals that their performance improvement over memory-free methods is attributed to using buffer exemplars to enhance knowledge retention. In particular, DER++ benefits from its training trajectory representation within a functional $L^2$ Hilbert space, preserving past knowledge. In contrast, LwF exhibits the lowest performance, primarily due to its lack of exemplar storage, which limits its ability to retain prior knowledge.

Notably, the LFL and the LFL+ leverage a stepwise freezing strategy to consolidate knowledge from previous tasks while facilitating adaptation to new ones. With this novel training method, the LFL+ outperforms most memory-based approaches, including Coil, iCaRL, and DER++, despite not having access to previous exemplars. However, despite these improvements, Fig. 2 shows a persistent performance gap between CL methods and the upper bound achieved by joint training. This underscores a fundamental challenge in CL: existing methodologies struggle to achieve optimal classification accuracy, necessitating more efficient knowledge retention and transfer strategies.

Furthermore, the performance of all methods declines noticeably as the number of new classes increases. This trend highlights that methods with higher PS scores are better suited for the CIL scenario. As shown in Table 2 for CIFAR-100, the performance of memory-based approaches improves proportionally with buffer size, suggesting that larger buffers can be leveraged to boost overall accuracy, albeit at the cost of increased computational and memory overhead. The evolution of the AF and I metrics across varying buffer sizes further reveals that Neural Networks in memory-based methods rely heavily on stored samples.

Additional results, including ACC, AF, I, and PS metrics for Tiny-ImageNet and ImageNet-1000, are provided in the Appendix. We also include experiments under the CIL setting with varying task granularities for each dataset to assess the impact of task partitioning, as well as an ablation study of each step's contribution for both the LFL and the LFL+, which are likewise detailed in the Appendix. We also provide a comparison based on TIL scenarios and method-wise in the Appendix. Additional metrics, including BWT and FWT, are reported in the Appendix.

## 4.2 MEMORY-WISE COMPARISON

Notably, the LFL+ achieves substantial improvements, comparable to MEMO, despite requiring 2.53% of DyTox's model buffer as shown in Table 3. This contrast underscores a fundamental issue in CL evaluation: the unfair comparison of methods with vastly different memory footprints. This is

Table 3: Memory usage comparison across methods (MB).

| Metric | LFL | LFL+ | EWC | SI | LwF | PODNet | BiC | GEM | iCaRL | Coil | DER++ | MEMO | DyTox |
|--------|-----|------|-----|-----|-----|--------|-----|-----|-------|------|-------|------|-------|
| Model | 44.68 | 46.24 | 44.68 | 44.68 | 44.68 | 44.68 | 44.68 | 44.68 | 44.68 | 44.68 | 44.68 | 682.40 | 1832.51 |
| Memory | 0.00 | 0.00 | 0.00 | 0.00 | 0.00 | 3010.56 | 3010.56 | 3010.56 | 3010.56 | 3010.56 | 3010.56 | 3010.56 | 3010.56 |
| Total | 44.68 | 46.24 | 44.68 | 44.68 | 44.68 | 3055.24 | 3055.24 | 3055.24 | 3055.24 | 3055.24 | 3055.24 | 3692.96 | 3055.24 |

primarily because memory-based methods and dynamic architectures inherently allocate additional memory and model resources, providing a clear advantage over methods such as LwF, EWC, LFL, and LFL+, which do not utilize exemplar buffers.

As discussed in the criteria for a method to be CL in the introduction section, a core principle of CL is the strict absence of access to past task data, which challenges the validity of current evaluation frameworks. To mitigate this unfair comparison, the authors in (Zhou et al., 2024b) proposed increasing the exemplar capacity for memory-based approaches. However, this adjustment is incompatible with LwF, EWC, SI, PNN, LFL, and LFL+ methods, which fundamentally do not store exemplars. Additionally, even for memory-based methods like iCaRL and Coil, simply expanding the exemplar set does not sufficiently bridge the gap caused by the higher memory requirements of dynamic architectures. These findings underscore the need for a more equitable benchmarking framework that systematically accounts for variations in memory consumption and data access constraints. Without such adjustments, existing evaluation paradigms may lead to misleading conclusions about the effectiveness of different CL approaches.

### 4.3 TRAINING TIME COMPARISON

Comparative training time benchmarks for each method are presented in Fig. 3 across all three datasets. The results show that memory-free methods, particularly the LFL and the LFL+, require significantly less training time compared to memory-based approaches. Among all, dynamic architecture-based methods are the most time-consuming. We attribute this to the repeated storage and reuse of parameters, along with frequent structural modifications during training.

The results also indicate that the CIL scenario consistently requires more training time than the TIL scenario, which is expected due to the added complexity of learning to classify across all previously seen classes in CIL, as opposed to task-specific classification in TIL. Furthermore, large-scale datasets such as ImageNet-1000 and Tiny-ImageNet, which contain a higher number of classes, naturally lead to longer training times. This underscores the importance of carefully designing task splits when working with large-scale datasets in CL.

## 5 CONCLUSION

We introduced the LFL and its enhanced variant, the LFL+. The LFL employs KD to mitigate Catastrophic Forgetting by preserving prior knowledge through soft-target supervision. Building upon this foundation, LFL+ integrates an under-complete Auto-Encoder to retain essential feature representations. LFL+ ensures knowledge retention, bias minimization, and stepwise freezing and fine-tuning for incremental learning.

In addition, we propose the Plasticity-Stability ratio to improve the evaluation of CL models. Extensive benchmarking demonstrates that the LFL and the LFL+ achieve an effective balance between learning new tasks and preserving performance on previous ones. Their performance, exhibited by memory-free CL frameworks, represents a scalable and efficient alternative to conventional approaches.

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

## A  APPENDIX

The Appendix mainly contains additional materials and experiments that cannot be reported due to the page limit, which is organized as follows:

- The Evaluation Protocol section B outlines CL's primary scenarios used in this paper.

- The Evaluation Metrics section C provides the mathematical definitions of the metrics used in this study.

- The Evaluation of Memory Setup section D outlines storing exemplars and model parameters used for analyzing the Table 3.

- The Additional Results section E presents a comparison based on the TIL scenario and method-wise comparison. In addition, extended evaluations under the CIL scenario, including further dataset results, experiments with varying task granularity, and additional evaluation metrics for TIL, such as FWT and BWT.

- The Ablation Study of LFL and LFL+, presented in Section F, examines the weight-pruning behavior of LwF and LFL to assess parameter utilization. In addition, this section analyzes the contribution of each step in LFL and LFL+ to the overall performance.

- The Pseudo-code section G presents the algorithms for both LFL and LFL+.

- The Implementation details of iCaRL section H describe how the method was adapted for the TIL scenario, given that its original design specifically targets the CIL setting.

- The Hyperparameter section I provides details about the best hyperparameters selected, as well as a comprehensive list of all parameter combinations that were evaluated.

Table 4: Methods characteristics. NM: No-Memory, D: Dynamic Network, R: Regularization.

| | LFL | LFL+ | LwF | EWC | SI | PNN | GEM | BiC | PODNet | DyTox | Coil | iCaRL | DER++ | MEMO |
|---|---|---|---|---|---|---|---|---|---|---|---|---|---|---|
| NM | ✓ | ✓ | ✓ | ✓ | ✓ | ✓ | ✗ | ✗ | ✗ | ✗ | ✗ | ✗ | ✗ | ✗ |
| D | - | - | - | - | - | ✓ | - | - | - | ✓ | - | - | - | ✓ |
| R | - | - | - | ✓ | ✓ | - | - | - | - | - | - | - | - | - |
| KD | ✓ | ✓ | ✓ | - | - | - | - | ✓ | ✓ | - | ✓ | ✓ | ✓ | - |

## B  Evaluation Protocol

The two main experimental scenarios typically used to evaluate the performance of methods are the following:

- **Task Incremental Learning (TIL):** In TIL, the training data is divided into multiple tasks, each with a unique set of classes. The crucial aspect of TIL is that the model is provided with information about which task it is handling during training and testing. This allows the model to use the computational graph corresponding to each task. For example, if the model is trained to classify images of animals and vehicles, the task label information is also provided for testing on a new image; thus, the network's classification output for the corresponding task will be calculated. This knowledge simplifies the inference task, as the model does not need to consider all possible classes simultaneously (Wickramasinghe et al., 2024; Vahedifar et al., 2025).

- **Class Incremental Learning (CIL):** In CIL, the model is also trained on different tasks, but is not told which task a new sample belongs to during testing. Instead, regardless of the task, the model needs to respond to all the classes it has encountered. This makes CIL more challenging than TIL, as the model must infer the correct class without task-related information. For instance, after training a model to recognize animals and vehicles separately, CIL would test the model on all classes simultaneously (animals and vehicles) without informing the model whether it is currently classifying an animal or a vehicle (Qu et al., 2024; Zhou et al., 2024a).

## C  Evaluation Metrices

When CL methods trained for task $T_k$ on $D_k$, its accuracy on all tasks is measured using the corresponding test sets, leading to a matrix $A \in \mathbb{R}^{T \times T}$ containing the accuracies on all $T$ tasks, i.e., $A_{i,j}$ denotes the accuracy of the model on task $T_j$ after trained completely on task $T_i$.

**Average Accuracy (ACC)** (Lopez-Paz & Ranzato, 2017): This metric assesses the overall performance of the CL method after completing the training of all $T$ tasks.

$$\text{ACC}_T = \frac{1}{T} \sum_{k=1}^{T} A_{T,k}. \tag{15}$$

**Forward transfer (FWT)** (Lopez-Paz & Ranzato, 2017): which is the influence that training on a $k$-th task has, on average, on the performance of the model on the next task:

$$\text{FWT}_T = \frac{1}{T-1} \sum_{k=2}^{T} (A_{k-1,k} - b_k) \tag{16}$$

Here, $b_k$ is the classification accuracy of a randomly initialized reference model for the $k$-th task.

**Backward transfer (BWT)** (Lopez-Paz & Ranzato, 2017): This metric evaluates the average influence of learning the $T$-th task on previous tasks:

$$\text{BWT}_T = \frac{1}{T-1} \sum_{k=1}^{T-1} (A_{T,k} - A_{k,k}). \tag{17}$$

Table 5: ACC Performance of CL methods for TIL scenario on CIFAR-100, Tiny-ImageNet, and ImageNet-1000 averaged over ten runs. The symbol "-" indicates experiments not conducted due to incompatibilities.

| | Method | CIFAR-100 | Tiny-ImageNet | ImageNet-1000 |
|---|---|---|---|---|
| **No Buffer** | LB | 58.41 ± 4.17 | 28.92 ± 2.978 | 21.54 ± 1.58 |
| | **LFL** | 83.45 ± 1.25 | 49.77 ± 5.38 | 46.19 ± 3.55 |
| | **LFL+** | 92.68 ± 2.45 | 58.21 ± 4.10 | 51.13 ± 3.19 |
| | LwF | 62.86 ± 3.50 | 25.85 ± 1.59 | 26.07 ± 2.59 |
| | SI | 67.13 ± 6.25 | 31.45 ± 4.87 | 27.79 ± 6.01 |
| | EWC | 60.59 ± 1.39 | 27.19 ± 3.45 | 23.22 ± 4.17 |
| | PNN | 91.26 ± 1.79 | 67.84 ± 2.91 | 52.13 ± 3.19 |
| **200** | GEM | 83.89 ± 3.96 | 46.32 ± 2.87 | - |
| | BiC | 81.09 ± 2.81 | 43.67 ± 2.92 | 39.47 ± 2.44 |
| | Coil | 77.99 ± 3.92 | 42.62 ± 2.81 | 35.89 ± 4.63 |
| | iCaRL | 81.14 ± 2.13 | 45.19 ± 2.44 | 37.27 ± 3.44 |
| | DER++ | 84.45 ± 2.60 | 51.50 ± 3.64 | 47.27 ± 2.78 |
| | PODNet | 78.63 ± 3.31 | 41.85 ± 4.12 | 41.19 ± 6.91 |
| | MEMO | 84.63 ± 1.99 | 53.14 ± 3.29 | 48.76 ± 4.63 |
| | DyTox | 85.63 ± 2.44 | 48.41 ± 3.21 | 48.83 ± 3.91 |
| **500** | GEM | 89.34 ± 4.69 | 50.15 ± 3.84 | - |
| | BiC | 87.14 ± 2.13 | 47.52 ± 3.24 | 43.27 ± 2.89 |
| | Coil | 81.25 ± 4.01 | 43.70 ± 1.97 | 40.19 ± 6.20 |
| | iCaRL | 86.93 ± 3.61 | 48.21 ± 3.76 | 39.44 ± 2.91 |
| | DER++ | 88.45 ± 3.59 | 53.61 ± 4.11 | 49.27 ± 3.09 |
| | PODNet | 81.99 ± 4.17 | 44.72 ± 3.96 | 43.19 ± 4.63 |
| | MEMO | 89.01 ± 4.13 | 53.75 ± 3.33 | 50.19 ± 3.63 |
| | DyTox | 90.55 ± 3.36 | 52.39 ± 2.95 | 50.21 ± 2.17 |
| **5120** | GEM | 90.86 ± 4.20 | 53.27 ± 3.61 | - |
| | BiC | 89.14 ± 3.91 | 50.73 ± 3.51 | 47.27 ± 4.21 |
| | Coil | 86.27 ± 3.33 | 46.21 ± 2.32 | 44.19 ± 4.63 |
| | iCaRL | 89.26 ± 3.22 | 49.03 ± 5.01 | 44.03 ± 2.97 |
| | DER++ | 90.45 ± 3.61 | 56.31 ± 4.96 | 53.27 ± 2.79 |
| | PODNet | 84.63 ± 5.95 | 47.15 ± 4.85 | 45.76 ± 3.63 |
| | MEMO | 90.87 ± 3.41 | 58.19 ± 3.85 | 54.19 ± 2.63 |
| | DyTox | 91.63 ± 2.14 | 55.84 ± 2.87 | 54.18 ± 2.09 |
| **—** | UB | 97.4 ± 0.12 | 89.26 ± 1.40 | 84.26 ± 1.26 |

**Average forgetting (AF)** (Chaudhry et al., 2018): It measures how much knowledge has been forgotten across the first $T-1$ tasks in TIL (or classes in CIL):

$$\text{AF}_T = \frac{1}{T-1} \sum_{j=1}^{T-1} f_T^j, \tag{18}$$

$$f_T^j = \max_{i \in \{1,\dots,T-1\}} A_{i,j} - A_{T,j}, \quad \forall j < T. \tag{19}$$

**Intransience measure (I)** (Chaudhry et al., 2018): It measures the impact on the model's accuracy when trained in a CL manner compared to training it using the typical batch learning:

$$I_T = A_T^* - A_{T,T}. \tag{20}$$

Here, $A_T^*$ denotes the model's accuracy if it were trained on the dataset $D = \cup_{t=1}^T D_t$.

To summarize, for a CL method, the higher the ACC, FWT, BWT, and PS, and the lower the AF and I in the trained models, the better the method is at combating Catastrophic Forgetting (Díaz-Rodríguez et al., 2018). If two models have similar ACC, the one with a larger PS, BWT, and/or FWT is preferable. Notably, Backward transfer for the first task and forward transfer for the last task are meaningless (Lopez-Paz & Ranzato, 2017).

Table 6: Forward Transfer (FWT) and Backward Transfer (BWT) performance of CL methods for TIL scenario averaged over ten runs. The symbol "-" indicates experiments not conducted due to incompatibilities.

| | Method | CIFAR-100 FWT ↑ | CIFAR-100 BWT ↑ | Tiny-ImageNet FWT ↑ | Tiny-ImageNet BWT ↑ | ImageNet-1000 FWT ↑ | ImageNet-1000 BWT ↑ |
|---|---|---|---|---|---|---|---|
| **No Buffer** | LB | -3.06 ± 2.90 | -55.39 ± 10.12 | -8.21 ± 3.74 | -64.61 ± 14.98 | -11.09 ± 4.01 | -79.24 ± 12.12 |
| | **LFL** | 8.51 ± 4.90 | -9.22 ± 8.49 | 8.02 ± 5.13 | -14.73 ± 9.41 | 6.51 ± 5.32 | -23.62 ± 8.57 |
| | **LFL+** | 13.98 ± 7.29 | -0.35 ± 7.45 | 9.41 ± 6.71 | -1.35 ± 5.56 | 7.31 ± 5.02 | -5.19 ± 7.16 |
| | LwF | 1.39 ± 4.06 | -39.69 ± 10.25 | 0.97 ± 7.26 | -41.21 ± 8.54 | 0.63 ± 5.12 | -58.59 ± 10.28 |
| | SI | -1.50 ± 5.27 | -45.78 ± 8.64 | -4.83 ± 6.15 | -52.97 ± 11.23 | -6.64 ± 4.20 | -63.76 ± 8.89 |
| | EWC | -4.44 ± 4.18 | -31.64 ± 8.37 | -6.28 ± 5.91 | -41.85 ± 9.64 | -7.51 ± 4.02 | -54.13 ± 11.41 |
| | PNN | - | - | - | - | - | - |
| **200** | GEM | 1.26 ± 3.08 | -9.61 ± 11.60 | 0.83 ± 4.15 | -12.84 ± 8.92 | - | - |
| | DyTox | 5.16 ± 1.97 | -0.25 ± 9.56 | 3.87 ± 2.65 | -3.96 ± 7.42 | 2.89 ± 3.74 | -5.88 ± 9.61 |
| | PODNet | 0.16 ± 1.41 | -9.57 ± 4.89 | -2.15 ± 3.87 | -13.91 ± 6.73 | -11.89 ± 4.88 | -18.35 ± 9.04 |
| | BiC | -2.76 ± 2.00 | -1.02 ± 10.45 | -4.85 ± 2.89 | -6.21 ± 8.15 | -7.94 ± 3.19 | -9.11 ± 7.42 |
| | Coil | -1.26 ± 2.22 | -6.40 ± 12.67 | -4.17 ± 3.47 | -9.78 ± 6.31 | -6.39 ± 4.30 | -14.36 ± 10.03 |
| | iCaRL | - | -2.72 ± 10.20 | - | -7.25 ± 5.01 | - | -10.01 ± 4.14 |
| | DER++ | -0.69 ± 1.86 | -8.59 ± 3.32 | -2.50 ± 5.13 | -14.10 ± 13.21 | -5.88 ± 6.90 | -15.16 ± 10.21 |
| | MEMO | -0.09 ± 2.09 | -2.39 ± 6.00 | -1.95 ± 3.75 | -3.82 ± 7.77 | -3.69 ± 2.04 | -5.17 ± 4.02 |
| **500** | GEM | 1.59 ± 3.26 | -7.31 ± 0.91 | 1.12 ± 4.58 | -9.73 ± 5.47 | - | - |
| | DyTox | 8.02 ± 4.02 | -1.36 ± 3.75 | 6.28 ± 5.17 | -4.85 ± 2.91 | 5.15 ± 6.53 | -7.10 ± 1.41 |
| | PODNet | 1.02 ± 0.93 | -3.44 ± 2.10 | -1.25 ± 4.87 | -5.21 ± 6.94 | -7.15 ± 7.13 | -6.35 ± 8.20 |
| | BiC | -1.59 ± 3.26 | -3.35 ± 0.31 | -2.87 ± 4.12 | -8.47 ± 5.69 | -4.53 ± 3.65 | -14.56 ± 1.44 |
| | Coil | -0.27 ± 2.80 | -4.13 ± 11.34 | -1.57 ± 6.30 | -7.96 ± 9.51 | -4.23 ± 8.35 | -11.16 ± 8.35 |
| | iCaRL | - | -5.71 ± 1.10 | - | -9.49 ± 3.53 | - | -13.06 ± 4.55 |
| | DER++ | 1.90 ± 2.07 | -2.38 ± 1.46 | -0.11 ± 2.17 | -4.50 ± 5.81 | -1.29 ± 3.00 | -5.66 ± 7.15 |
| | MEMO | 0.29 ± 2.23 | -2.36 ± 2.00 | -0.65 ± 4.44 | -4.36 ± 3.27 | -0.97 ± 2.56 | -5.10 ± 4.46 |
| **5120** | GEM | 2.51 ± 2.53 | -4.94 ± 10.39 | 1.98 ± 3.74 | -7.15 ± 8.23 | - | - |
| | DyTox | 11.59 ± 4.44 | -0.56 ± 11.23 | 9.21 ± 5.32 | -2.94 ± 7.41 | 8.98 ± 4.84 | -5.10 ± 5.90 |
| | PODNet | 2.59 ± 4.44 | -2.32 ± 12.00 | 0.97 ± 5.73 | -3.18 ± 9.64 | -1.98 ± 4.84 | -3.85 ± 11.52 |
| | BiC | 0.51 ± 3.53 | -1.44 ± 10.89 | -0.82 ± 4.21 | -5.29 ± 8.76 | -0.18 ± 4.27 | -8.31 ± 11.45 |
| | Coil | 2.71 ± 1.05 | -1.24 ± 11.44 | 0.16 ± 3.46 | -5.24 ± 5.37 | -1.23 ± 2.67 | -7.55 ± 4.90 |
| | iCaRL | - | -4.94 ± 4.10 | - | -8.64 ± 3.47 | - | -9.87 ± 3.81 |
| | DER++ | 3.71 ± 2.00 | -1.33 ± 9.14 | 2.76 ± 1.11 | -2.98 ± 6.44 | 1.23 ± 0.63 | -3.10 ± 8.15 |
| | MEMO | 10.25 ± 6.63 | -1.56 ± 9.39 | 8.19 ± 4.26 | -2.18 ± 5.71 | 7.76 ± 5.13 | -4.18 ± 4.00 |
| **—** | UB | - | - | - | - | - | - |

# D EVALUATION MEMORY SETUP

We evaluate the total memory cost of ResNet-18 on ImageNet-1000 by summing the memory required for storing exemplars and model parameters. Each ImageNet-1000 exemplar requires $3 \times 224 \times 224 = 150,528$ bytes. With 20,000 exemplars, the total memory footprint for stored images is approximately $3,010.56$ MB across exemplar-based methods. The model memory varies based on the number of model parameters. Each parameter is stored as a 32-bit float (4 bytes). Most methods, including LWF, iCaRL, Coil, DER++, and LFL, contain 11.17 Million (M) parameters, hence requiring $44.68$ MB, while LFL+ has 11.56 M parameters (i.e. $46.24$ MB). DyTox is the most memory-intensive, requiring $1832.51$ MB.

# E ADDITIONAL RESULTS

In this section, we present a comparison of CIL performance across 20 tasks for all three datasets. Additionally, we provide a method-wise comparison to further highlight the differences among the evaluated approaches.

## E.1 COMPARISON BASED ON THE TIL SCENARIO

We evaluate CL methods across multiple datasets by structuring learning into $T = 10$ sequential tasks, where each data instance is encountered only once during training. As shown in Table 5 and

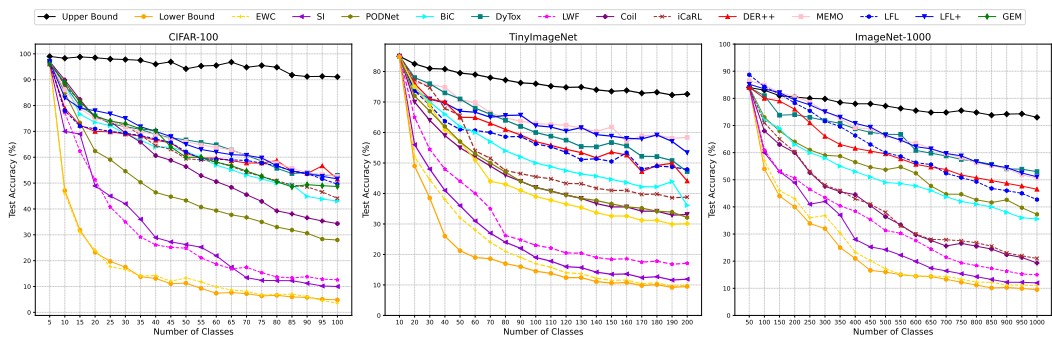

Figure 4: ACC evaluation comparison for the CIL for 20 tasks for each dataset.

Table 6, performance trends across most methods remain consistent across different datasets. The TIL scenario is generally easier for CL methods, as task boundaries are known during inference. In TIL, models are only required to classify among the classes of the current task. In contrast, CIL requires models to classify among all classes encountered so far, without access to task identity, making it a more challenging and realistic setting. The consistently higher accuracy observed in the TIL scenario compared to CIL further confirms this distinction.

A study on varying memory sizes in the TIL scenario (200, 500, and 5120 exemplars) shows that memory-based methods do benefit from larger exemplar sets. However, when comparing this to results in the CIL scenario, it becomes evident that increasing buffer size provides greater gains in CIL than in TIL. Specifically, the performance improvement from 500 to 5120 exemplars in TIL is relatively modest, whereas the increase in memory consumption is substantial. This suggests that memory-based methods are more sensitive to buffer size in the CIL setting, where the need to distinguish between all previously learned classes makes the effective use of stored samples more critical.

### E.2 METHOD-WISE COMPARISON

We report the results of the CIL scenario for each dataset under a configuration of 20 tasks, as illustrated in Fig. 4. In this setting, the 100, 200, and 1000 classes from CIFAR-100, Tiny-ImageNet, and ImageNet-1000 correspond to 5, 10, and 50 incremental classes per task, respectively. This setup allows us to analyze how both the number of tasks and the number of classes per task influence overall performance. Fig. 2 demonstrates that, performance consistently declines as the number of tasks increases. This observation raises an important question: given a fixed number of classes, what is the optimal task partitioning strategy to maintain the best performance?

Furthermore, the granularity of task division significantly affects ACC, especially as the number of tasks grows. This suggests that many existing CL methods implicitly depend on grouping a large number of classes within a single task. Consequently, the feature extractor becomes critical, since learning multiple classes simultaneously requires extracting a richer set of features. In addition, we observe bias in the final classification layer. Specifically, the last layer tends to favor newly introduced classes, as it is updated with their data, which in turn makes it less effective at preserving knowledge of previously learned classes. These findings suggest the need for memory-free methods that more effectively coordinate the feature extractor and classification head.

Additionally, as shown in Table 2 and illustrated in Fig. 4, our findings indicate that methods achieving higher final performance generally exhibit lower levels of forgetting. This correlation arises because the final performance directly affects the second term in the forgetting calculation, underscoring the intrinsic relationship between these metrics. While helping with stability, regularization terms can reduce the model's ability to adapt to new tasks (plasticity), leading to poorer PS performance. This highlights the importance of the PS metric.

Table 7: ACC, AF, I, and PS measure on Tiny-ImageNet for CIL scenario with 20 incremental classes.

| Method | ACC↑ | AF↓ | I↓ | PS↑ |
|--------|------|-----|-----|-----|
| MEMO | 71.03 | 0.5191 | -0.0007 | 0.8000 |
| DyTox | 70.27 | 0.3800 | -0.0015 | 0.7200 |
| **LFL+** | 68.86 | 0.4866 | -0.0027 | 0.8811 |
| DER++ | 67.21 | 0.4672 | 0.0003 | 0.4500 |
| **LFL** | 64.76 | 0.5763 | -0.0007 | 0.6933 |
| BiC | 62.12 | 0.4200 | 0.0920 | 0.5500 |
| iCaRL | 61.57 | 0.5300 | -0.0024 | 0.4233 |
| GEM | 60.56 | 0.3500 | 0.0080 | 0.5800 |
| Coil | 56.49 | 0.5356 | -0.0002 | 0.3967 |
| PODNet | 55.81 | 0.5800 | 0.0240 | 0.4200 |
| LwF | 44.55 | 0.6195 | 0.0013 | 0.2233 |
| SI | 34.28 | 0.7200 | 0.0150 | 0.2800 |
| EWC | 26.70 | 1.0000 | -0.0080 | 0.1500 |

Table 8: ACC, AF, I, and PS measure on ImageNet-1000 for CIL scenario with 100 incremental classes.

| Method | ACC↑ | AF↓ | I↓ | PS↑ |
|--------|------|-----|-----|-----|
| DyTox | 69.91 | 0.3094 | 0.5754 | 0.5904 |
| MEMO | 69.62 | 0.4087 | 0.5541 | 0.5967 |
| **LFL+** | 66.36 | 0.3807 | 0.1307 | 0.5767 |
| DER++ | 64.80 | 0.3033 | 0.2338 | 0.5933 |
| **LFL** | 62.96 | 0.4880 | 0.5491 | 0.3067 |
| BiC | 61.36 | 0.4290 | 0.5754 | 0.3850 |
| PODNet | 60.86 | 0.3423 | 0.0115 | 0.3701 |
| Coil | 54.50 | 0.7319 | -0.2014 | 0.3656 |
| iCaRL | 53.91 | 0.4924 | -0.2028 | 0.3344 |
| LwF | 41.95 | 0.8013 | 0.0920 | 0.1611 |
| SI | 35.61 | 0.7602 | -0.0726 | 0.1000 |
| EWC | 31.71 | 0.9056 | -0.0783 | 0.1000 |

By analyzing Tables 2, we can observe that the ACC across all methods tends to decrease slightly when moving from 10 to 5 incremental classes per task. This is expected as increasing the number of incremental steps generally increases task complexity. However, the LFL+ consistently achieves the highest ACC in both settings (66.97 for five incremental classes per task and 67.18 for ten incremental classes per task), demonstrating its effectiveness. The LFL+ maintains the highest plasticity in both cases (0.5252 for five incremental classes per task and 0.4287 for ten incremental classes per task).

For Tiny-ImageNet, ACC is presented in Fig. 2, with additional metrics, ACC, AF, I, and PS, summarized in Table 7. Similarly, for ImageNet-1000, ACC is illustrated in Fig. 2, while Table 8 summarizes the ACC, AF, I, and PS metrics. The performance trends across most methods align with those observed on CIFAR-100. However, on large-scale datasets, the performance of these methods deteriorates as the number of classes that must be learned in each task increases. These results suggest that when designing a CL framework, ACC should not be the sole evaluation metric. The PS metric is equally important, as it reflects the model's ability to acquire new knowledge (plasticity) while retaining previously learned information (stability).

In addition, Table 2 shows that memory buffer-based methods significantly improve performance as more examples are stored per task. However, this improvement comes at the cost of a substantially larger memory footprint, which becomes a limitation as the dataset size or number of tasks increases, posing challenges for real-world deployment. Furthermore, results from large-scale datasets in Table 7 and Table 8 indicate that dynamic architecture-based methods outperform others in accuracy.

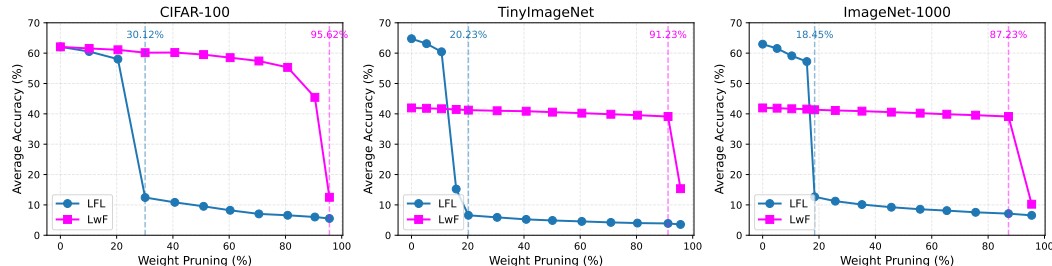

Figure 5: Weight pruning robustness comparison between LFL and LwF across three datasets in the CIL scenario. The dashed vertical lines indicate the critical pruning percentage where each method experiences significant accuracy degradation.

At the same time, LFL and LFL+ deliver comparable performance while requiring fewer resources, particularly in terms of memory usage.

Additionally, PS metric comparisons reveal that regularization-based methods such as EWC and SI perform worse, likely due to their strong reliance on anchoring training loss to past knowledge. In contrast, LFL and LFL+ demonstrate strong performance in the PS metric, on par with DyTox and MEMO (dynamic architecture-based approaches).

## F ABLATION STUDY OF THE LFL AND THE LFL+

In this section, we compare LFL and LwF under weight pruning to examine differences in weight utilization. Furthermore, we analyze the performance contribution of each step to the overall effectiveness of LFL and LFL+.

### F.1 WEIGHT PRUNING ANALYSIS OF LwF AND LFL

In this experiment, we analyze weight utilization in LFL and LwF under the common decomposition of the Neural Network parameters into $(\theta_s, \theta_t, \theta_{t+1})$. Our findings show that LFL exhibits early sensitivity to weight pruning, with performance dropping substantially at 18–30% pruning. In contrast, LwF remains remarkably robust, maintaining stable performance even under 87–95% pruning across all datasets, as illustrated in Fig. 5. This observation aligns with the Lottery Ticket Hypothesis (Frankle et al., 2020), suggesting that Neural Networks are heavily over-parameterized—and continual learning is no exception. The results in Fig. 5 further demonstrate that LFL's stepwise freezing strategy leverages redundancy in the network weights, attempting to preserve useful capacity rather than allowing these parameters to be overwritten or forgotten.

### F.2 THE LFL ABLATION STUDY ANALYSIS

The LFL ablation study in Table 9 demonstrates a consistent increase in accuracy with the addition of each proposed step. The "Task-specific Head Training" contributes significantly, leading to a 16.74% improvement. A substantial gain of 13.55% is observed with "Knowledge Distillation,". This progressive improvement highlights that Steps 2 and 3 contribute most to the LFL framework.

### F.3 LFL+ ABLATION STUDY ANALYSIS

Table 10 presents the ablation study for LFL+, an enhanced version of LFL. The introduction of "Auto-Encoder Training" in Step 2 shows a significant jump of 22.18% to 38.77%, indicating its role in feature preservation. "Knowledge Distillation" (Step 4) contributes improvement of 12.74%. This suggests that steps 2, 3, and 4 play a significant role in the model's performance.

Table 9: Ablation study of each LFL step on CIFAR-100 (CIL scenario with 10 incremental classes, No Buffer).

| Method Configuration | $\Delta$(%) | ACC (%) |
|---|---|---|
| Step 1: Basic Training on $T_t$ | - | 22.18 |
| Step 2: Task-specific Head Training | +16.74 | 38.92 |
| Step 3: Knowledge Distillation | +13.55 | 52.47 |
| Step 4: Fine-tuning and Dual Logit Targets (LFL) | +10.74 | 63.21 |

Table 10: Ablation study of each LFL+ step on CIFAR-100 (CIL scenario with 10 incremental classes, No Buffer).

| Method Configuration | $\Delta$(%) | ACC (%) |
|---|---|---|
| Step 1: Basic Training on $T_t$ | - | 22.18 |
| Step 2: Auto-Encoder Training | +16.59 | 38.77 |
| Step 3: Task-specific Head Training | +11.44 | 50.21 |
| Step 4: Knowledge Distillation | +12.74 | 62.95 |
| Step 5: Fine-tuning and Bias Correction (LFL+) | +4.23 | 67.18 |

## G   PSEUDO-CODE OF THE LFL AND THE LFL+

In this section, we present the pseudo-code for the LFL and the LFL+ to clarify the shared steps and highlight their key differences. As discussed in the main body of the paper, the LFL+ integrates an Auto-Encoder after the learning task $T_t$ to preserve the most informative features learned thus far. Overall, Steps 3 and 4 in LFL+, Algorithm 2, correspond to Steps 2 and 3 in the LFL, Algorithm 1, with the primary distinction being the use of unbiased logits in LFL+ to improve model performance.

## H   IMPLEMENTATION DETAILS OF iCARL

This section outlines our adaptation of iCaRL (Rebuffi et al., 2017) for the TIL setting, which builds upon its original proposal for CIL. A key modification involves refining its classification mechanism.

Conventionally, iCaRL determines a label $y^*$ by identifying the class whose average exemplar feature vector is most proximate to the input example's feature vector. Specifically, given $\overline{\mathbf{y}}$ as the average feature vector of exemplars for class $y$ and $\phi(\mathbf{x})$ as the feature vector derived from example $\mathbf{x}$, iCaRL's prediction is formulated as:

$$y^* = \underset{y=1,\ldots,t}{\operatorname{argmin}} \|\phi(\mathbf{x}) - \overline{\mathbf{y}}\|_2. \tag{21}$$

Our modified approach, however, casts iCaRL's network response, $h(\mathbf{x})$, in terms of the negative Euclidean distance to the tensor of average feature vectors for all classes, $\boldsymbol{\Phi}$. This yields:

$$h(\mathbf{x}) = -\|\phi(\mathbf{x}) - \boldsymbol{\Phi}\|_2. \tag{22}$$

It is pertinent to note that when considering the argmax of $h(\mathbf{x})$ in a CIL context, this formulation yields an identical prediction to that of Eq. 21.

Furthermore, it is noteworthy that iCaRL integrates a weight-decay regularization term, a crucial element for ensuring its competitive performance against other proposed approaches.

## I   HYPERPARAMETER SEARCH

We show the complete hyperparameter space in Table 11 and best hyperparameter combination that we chose for each method for the experiments in the main paper Table 12 and Table 13. We denote

---

**Algorithm 1** Less Forgetting Learning (LFL)

---

**Inputs:** Dataset $D_{t+1}(X_{t+1}, Y_{t+1})$ describing $T_{t+1}$, Parameters $\{\theta_s, \theta_t\}$ trained on $T_t$

**Step 1:** Train $\text{NN}^1$ with $\mathcal{L}^1$ from Eq. 1, calculate Soft Targets with Eq. 2

$$\text{NN}^1(X_t, \theta_s, \theta_t) \xrightarrow{Training} \text{NN}^1(O_t^1, \theta_s^*, \theta_t^*).$$

**Step 2:** Train $\text{NN}^2$ with $\mathcal{L}^2$ from Eq. 3

$$\text{NN}^2(X_{t+1}, \theta_s^*, \theta_{t+1}) \xrightarrow{Training} \text{NN}^2(O_{t+1}^2, \theta_s^*, \theta_{t+1}^*).$$

**Step 3:** Train $\text{NN}^3$ with $\mathcal{L}^3$ from Eq. 4

$$\text{NN}^3(X_{t+1}, \theta_s^*, \theta_t^*, \theta_{t+1}^*) \xrightarrow{Training}$$
$$\text{NN}^3(O_t^3, \theta_s^u, \theta_t^u); \quad \text{NN}^3(O_{t+1}^3, \theta_s^u, \theta_{t+1}^*)$$

**Step 4:** Compute new logits with Eq. 6, Train $\text{NN}^4$ with $\mathcal{L}^4$ from Eq. 7

$$\text{NN}^4(X_{t+1}, \theta_s^u, \theta_t^*, \theta_t^u, \theta_{t+1}^*) \xrightarrow{Training} \text{NN}^4(O_t^4, \theta_s^f, \theta_t^*);$$
$$\text{NN}^4(O_t'^4, \theta_s^f, \theta_t^f); \quad \text{NN}^4(O_{t+1}^4, \theta_s^f, \theta_{t+1}^f),$$

**Output:** Share Parameters $\{\theta_s^f\}$, Task Specfic Parameters $\{\theta_t^f, \theta_{t+1}^f\}$ for the next task

---

---

**Algorithm 2** LFL+

---

**Inputs:** Dataset $D_{t+1}(X_{t+1}, Y_{t+1})$ describing $T_{t+1}$, Parameters $\{\theta_s, \theta_t\}$ trained on $T_t$

**Step 1:** Corresponding to the LFL's Step 1

**Step 2:** Train Auto-Encoder for Feature Preservation with Eq. 9

**Step 3-4:** Corresponding to the LFL's steps 2-3, respectively

**Step 5:** Adjust Logit for Bias Correction with Eq. 10, Train $\text{NN}^5$ with $\mathcal{L}_5$ from Eq. 13

**Output:** Share Parameters $\{\theta_s^f\}$, Task Specific Parameters $\{\theta_t^f, \theta_{t+1}^f\}$ for the next task

---

the learning rate with $lr$, weight decay with $wd$, and Adam optimizer hyperparameters for Auto-Encoder training with $lr_{adam}, \epsilon, \beta_1$, and $\beta_2$.

Table 11: Complete hyperparameter search space for all methods across datasets.

| Method | Scenario | Dataset | Buffer | Hyperparameters |
|---|---|---|---|---|
| LFL | CIL/TIL | CIFAR-100
Tiny-ImageNet
ImageNet-1000 | -
-
- | $lr$: [0.001, 0.01, 0.03, 0.1, 0.2, 0.3],    $p$: [2.0, 3.0, 4.0],    $\alpha$: [0.3, 0.5, 0.7]
$lr$: [0.001, 0.01, 0.03, 0.1, 0.2, 0.3],    $p$: [2.0, 3.0, 4.0],    $\alpha$: [0.3, 0.5, 0.7]
$lr$: [0.001, 0.01, 0.03, 0.1, 0.2, 0.3],    $p$: [2.0, 3.0, 4.0],    $\alpha$: [0.3, 0.5, 0.7] |
| LFL+ | CIL/TIL | CIFAR-100

Tiny-ImageNet

ImageNet-1000 | -

-

- | $lr$: [0.001, 0.01, 0.03, 0.1, 0.2, 0.3],   $p$: [2.0, 3.0, 4.0],   $\Omega$: [0.1, 0.3, 0.5, 1.0],   $\eta$: [0.3, 0.5, 0.7]
$lr_{adam}$: [0.0001, 0.001, 0.01],   $\beta_1$: [0.9],   $\beta_2$: [0.999],   $\epsilon$: [1e-8]
$lr$: [0.001, 0.01, 0.03, 0.1, 0.2, 0.3],   $p$: [2.0, 3.0, 4.0],   $\Omega$: [0.1, 0.3, 0.5, 1.0],   $\eta$: [0.3, 0.5, 0.7]
$lr_{adam}$: [0.0001, 0.001, 0.01],   $\beta_1$: [0.9],   $\beta_2$: [0.999],   $\epsilon$: [1e-8]
$lr$: [0.001, 0.01, 0.03, 0.1, 0.2, 0.3],   $p$: [2.0, 3.0, 4.0],   $\Omega$: [0.1, 0.3, 0.5, 1.0],   $\eta$: [0.3, 0.5, 0.7]
$lr_{adam}$: [0.0001, 0.001, 0.01],   $\beta_1$: [0.9],   $\beta_2$: [0.999],   $\epsilon$: [1e-8] |
| LwF | CIL/TIL | CIFAR-100
Tiny-ImageNet
ImageNet-1000 | -
-
- | $lr$: [0.001, 0.01, 0.03, 0.1, 0.2, 0.3],   $\alpha$: [0.3, 0.5, 1.0, 3.0],   $T$: [2.0, 4.0],   $wd$: [1e-5, 5e-5]
$lr$: [0.001, 0.01, 0.03, 0.1, 0.2, 0.3],   $\alpha$: [0.3, 0.5, 1.0, 3.0],   $T$: [2.0, 4.0],   $wd$: [1e-5, 5e-5]
$lr$: [0.001, 0.01, 0.03, 0.1, 0.2, 0.3],   $\alpha$: [0.3, 0.5, 1.0, 3.0],   $T$: [2.0, 4.0],   $wd$: [1e-5, 5e-5] |
| EWC | CIL/TIL | CIFAR-100
Tiny-ImageNet
ImageNet-1000 | -
-
- | $lr$: [0.001, 0.01, 0.03, 0.1, 0.2, 0.3],   $\lambda$: [10, 25, 30, 90, 100],   $\gamma$: [0.9, 0.95, 1.0]
$lr$: [0.001, 0.01, 0.03, 0.1, 0.2, 0.3],   $\lambda$: [10, 25, 30, 90, 100],   $\gamma$: [0.9, 0.95, 1.0]
$lr$: [0.001, 0.01, 0.03, 0.1, 0.2, 0.3],   $\lambda$: [10, 25, 30, 90, 100],   $\gamma$: [0.9, 0.95, 1.0] |
| SI | CIL/TIL | CIFAR-100
Tiny-ImageNet
ImageNet-1000 | -
-
- | $lr$: [0.001, 0.01, 0.03, 0.1, 0.2, 0.3],   $c$: [0.3, 0.5, 0.7, 1.0],   $\xi$: [0.9, 1.0]
$lr$: [0.001, 0.01, 0.03, 0.1, 0.2, 0.3],   $c$: [0.3, 0.5, 0.7, 1.0],   $\xi$: [0.9, 1.0]
$lr$: [0.001, 0.01, 0.03, 0.1, 0.2, 0.3],   $c$: [0.3, 0.5, 0.7, 1.0],   $\xi$: [0.9, 1.0] |
| PNN | CIL/TIL | CIFAR-100
Tiny-ImageNet
ImageNet-1000 | -
-
- | $lr$: [0.001, 0.01, 0.03, 0.1, 0.2, 0.3]
$lr$: [0.001, 0.01, 0.03, 0.1, 0.2, 0.3]
$lr$: [0.001, 0.01, 0.03, 0.1, 0.2, 0.3] |
| GEM | CIL | CIFAR-100
Tiny-ImageNet
ImageNet-1000 | [1k, 2k]
[1k, 2k]
[10k, 20k] | $lr$: [0.001, 0.01, 0.03, 0.1, 0.2, 0.3],   $\gamma$: [0.1, 0.5, 1.0]
$lr$: [0.001, 0.01, 0.03, 0.1, 0.2, 0.3],   $\gamma$: [0.1, 0.5, 1.0]
$lr$: [0.001, 0.01, 0.03, 0.1, 0.2, 0.3],   $\gamma$: [0.1, 0.5, 1.0] |
| GEM | TIL | CIFAR-100
Tiny-ImageNet
ImageNet-1000 | [200, 500, 5120]
[200, 500, 5120]
[200, 500, 5120] | $lr$: [0.001, 0.01, 0.03, 0.1, 0.2, 0.3],   $\gamma$: [0.1, 0.5, 1.0]
$lr$: [0.001, 0.01, 0.03, 0.1, 0.2, 0.3],   $\gamma$: [0.1, 0.5, 1.0]
$lr$: [0.001, 0.01, 0.03, 0.1, 0.2, 0.3],   $\gamma$: [0.1, 0.5, 1.0] |
| BiC | CIL | CIFAR-100
Tiny-ImageNet
ImageNet-1000 | [1k, 2k]
[1k, 2k]
[10k, 20k] | $lr$: [0.001, 0.01, 0.03, 0.1, 0.2, 0.3]
$lr$: [0.001, 0.01, 0.03, 0.1, 0.2, 0.3]
$lr$: [0.001, 0.01, 0.03, 0.1, 0.2, 0.3] |
| BiC | TIL | CIFAR-100
Tiny-ImageNet
ImageNet-1000 | [200, 500, 5120]
[200, 500, 5120]
[200, 500, 5120] | $lr$: [0.001, 0.01, 0.03, 0.1, 0.2, 0.3]
$lr$: [0.001, 0.01, 0.03, 0.1, 0.2, 0.3]
$lr$: [0.001, 0.01, 0.03, 0.1, 0.2, 0.3] |
| PODNet | CIL | CIFAR-100
Tiny-ImageNet
ImageNet-1000 | [1k, 2k]
[1k, 2k]
[10k, 20k] | $lr$: [0.001, 0.01, 0.03, 0.1, 0.2, 0.3],   $\lambda_{pod}$: [0.1, 0.3, 0.5, 1.0, 3.0]
$lr$: [0.001, 0.01, 0.03, 0.1, 0.2, 0.3],   $\lambda_{pod}$: [0.1, 0.3, 0.5, 1.0, 3.0]
$lr$: [0.001, 0.01, 0.03, 0.1, 0.2, 0.3],   $\lambda_{pod}$: [0.1, 0.3, 0.5, 1.0, 3.0] |
| PODNet | TIL | CIFAR-100
Tiny-ImageNet
ImageNet-1000 | [200, 500, 5120]
[200, 500, 5120]
[200, 500, 5120] | $lr$: [0.001, 0.01, 0.03, 0.1, 0.2, 0.3],   $\lambda_{pod}$: [0.1, 0.3, 0.5, 1.0, 3.0]
$lr$: [0.001, 0.01, 0.03, 0.1, 0.2, 0.3],   $\lambda_{pod}$: [0.1, 0.3, 0.5, 1.0, 3.0]
$lr$: [0.001, 0.01, 0.03, 0.1, 0.2, 0.3],   $\lambda_{pod}$: [0.1, 0.3, 0.5, 1.0, 3.0] |
| DyTox | CIL | CIFAR-100
Tiny-ImageNet
ImageNet-1000 | [1k, 2k]
[1k, 2k]
[10k, 20k] | $lr$: [0.001, 0.01, 0.03, 0.1, 0.2, 0.3]
$lr$: [0.001, 0.01, 0.03, 0.1, 0.2, 0.3]
$lr$: [0.001, 0.01, 0.03, 0.1, 0.2, 0.3] |
| DyTox | TIL | CIFAR-100
Tiny-ImageNet
ImageNet-1000 | [200, 500, 5120]
[200, 500, 5120]
[200, 500, 5120] | $lr$: [0.001, 0.01, 0.03, 0.1, 0.2, 0.3]
$lr$: [0.001, 0.01, 0.03, 0.1, 0.2, 0.3]
$lr$: [0.001, 0.01, 0.03, 0.1, 0.2, 0.3] |
| Coil | CIL | CIFAR-100
Tiny-ImageNet
ImageNet-1000 | [1k, 2k]
[1k, 2k]
[10k, 20k] | $lr$: [0.001, 0.01, 0.03, 0.1, 0.2, 0.3],   $\lambda_{coil}$: [0.1, 0.3, 0.5, 1.0]
$lr$: [0.001, 0.01, 0.03, 0.1, 0.2, 0.3],   $\lambda_{coil}$: [0.1, 0.3, 0.5, 1.0]
$lr$: [0.001, 0.01, 0.03, 0.1, 0.2, 0.3],   $\lambda_{coil}$: [0.1, 0.3, 0.5, 1.0] |
| Coil | TIL | CIFAR-100
Tiny-ImageNet
ImageNet-1000 | [200, 500, 5120]
[200, 500, 5120]
[200, 500, 5120] | $lr$: [0.001, 0.01, 0.03, 0.1, 0.2, 0.3],   $\lambda_{coil}$: [0.1, 0.3, 0.5, 1.0]
$lr$: [0.001, 0.01, 0.03, 0.1, 0.2, 0.3],   $\lambda_{coil}$: [0.1, 0.3, 0.5, 1.0]
$lr$: [0.001, 0.01, 0.03, 0.1, 0.2, 0.3],   $\lambda_{coil}$: [0.1, 0.3, 0.5, 1.0] |
| iCaRL | CIL | CIFAR-100
Tiny-ImageNet
ImageNet-1000 | [1k, 2k]
[1k, 2k]
[10k, 20k] | $lr$: [0.001, 0.01, 0.03, 0.1, 0.2, 0.3],   $wd$: [0, 1e-5, 5e-5, 1e-4]
$lr$: [0.001, 0.01, 0.03, 0.1, 0.2, 0.3],   $wd$: [0, 1e-5, 5e-5, 1e-4]
$lr$: [0.001, 0.01, 0.03, 0.1, 0.2, 0.3],   $wd$: [0, 1e-5, 5e-5, 1e-4] |
| iCaRL | TIL | CIFAR-100
Tiny-ImageNet
ImageNet-1000 | [200, 500, 5120]
[200, 500, 5120]
[200, 500, 5120] | $lr$: [0.001, 0.01, 0.03, 0.1, 0.2, 0.3],   $wd$: [0, 1e-5, 5e-5, 1e-4]
$lr$: [0.001, 0.01, 0.03, 0.1, 0.2, 0.3],   $wd$: [0, 1e-5, 5e-5, 1e-4]
$lr$: [0.001, 0.01, 0.03, 0.1, 0.2, 0.3],   $wd$: [0, 1e-5, 5e-5, 1e-4] |
| DER | CIL | CIFAR-100
Tiny-ImageNet
ImageNet-1000 | [1k, 2k]
[1k, 2k]
[10k, 20k] | $lr$: [0.001, 0.01, 0.03, 0.1, 0.2, 0.3],   $\alpha$: [0.1, 0.2, 0.3, 0.5, 1.0]
$lr$: [0.001, 0.01, 0.03, 0.1, 0.2, 0.3],   $\alpha$: [0.1, 0.2, 0.3, 0.5, 1.0]
$lr$: [0.001, 0.01, 0.03, 0.1, 0.2, 0.3],   $\alpha$: [0.1, 0.2, 0.3, 0.5, 1.0] |
| DER | TIL | CIFAR-100
Tiny-ImageNet
ImageNet-1000 | [200, 500, 5120]
[200, 500, 5120]
[200, 500, 5120] | $lr$: [0.001, 0.01, 0.03, 0.1, 0.2, 0.3],   $\alpha$: [0.1, 0.2, 0.3, 0.5, 1.0]
$lr$: [0.001, 0.01, 0.03, 0.1, 0.2, 0.3],   $\alpha$: [0.1, 0.2, 0.3, 0.5, 1.0]
$lr$: [0.001, 0.01, 0.03, 0.1, 0.2, 0.3],   $\alpha$: [0.1, 0.2, 0.3, 0.5, 1.0] |
| DER++ | CIL | CIFAR-100
Tiny-ImageNet
ImageNet-1000 | [1k, 2k]
[1k, 2k]
[10k, 20k] | $lr$: [0.001, 0.01, 0.03, 0.1, 0.2, 0.3],   $\alpha$: [0.1, 0.2, 0.3, 0.5, 1.0],   $\beta$: [0.5, 1.0]
$lr$: [0.001, 0.01, 0.03, 0.1, 0.2, 0.3],   $\alpha$: [0.1, 0.2, 0.3, 0.5, 1.0],   $\beta$: [0.5, 1.0]
$lr$: [0.001, 0.01, 0.03, 0.1, 0.2, 0.3],   $\alpha$: [0.1, 0.2, 0.3, 0.5, 1.0],   $\beta$: [0.5, 1.0] |
| DER++ | TIL | CIFAR-100
Tiny-ImageNet
ImageNet-1000 | [200, 500, 5120]
[200, 500, 5120]
[200, 500, 5120] | $lr$: [0.001, 0.01, 0.03, 0.1, 0.2, 0.3],   $\alpha$: [0.1, 0.2, 0.3, 0.5, 1.0],   $\beta$: [0.5, 1.0]
$lr$: [0.001, 0.01, 0.03, 0.1, 0.2, 0.3],   $\alpha$: [0.1, 0.2, 0.3, 0.5, 1.0],   $\beta$: [0.5, 1.0]
$lr$: [0.001, 0.01, 0.03, 0.1, 0.2, 0.3],   $\alpha$: [0.1, 0.2, 0.3, 0.5, 1.0],   $\beta$: [0.5, 1.0] |
| MEMO | CIL | CIFAR-100
Tiny-ImageNet
ImageNet-1000 | [1k, 2k]
[1k, 2k]
[10k, 20k] | $lr$: [0.001, 0.01, 0.03, 0.1, 0.2, 0.3]
$lr$: [0.001, 0.01, 0.03, 0.1, 0.2, 0.3]
$lr$: [0.001, 0.01, 0.03, 0.1, 0.2, 0.3] |
| MEMO | TIL | CIFAR-100
Tiny-ImageNet
ImageNet-1000 | [200, 500, 5120]
[200, 500, 5120]
[200, 500, 5120] | $lr$: [0.001, 0.01, 0.03, 0.1, 0.2, 0.3]
$lr$: [0.001, 0.01, 0.03, 0.1, 0.2, 0.3]
$lr$: [0.001, 0.01, 0.03, 0.1, 0.2, 0.3] |

Table 12: Selected best hyperparameters for all methods across datasets- Part I.

| Method | Scenario | Dataset | Buffer | Selected Hyperparameters |
|---|---|---|---|---|
| LFL | CIL/TIL | CIFAR-100 | - | $lr$: 0.1, $p$: 2.0, $\alpha$: 0.3 |
| | | Tiny-ImageNet | - | $lr$: 0.03, $p$: 2.0, $\alpha$: 0.3 |
| | | ImageNet-1000 | - | $lr$: 0.03, $p$: 2.0, $\alpha$: 0.5 |
| LFL+ | CIL/TIL | CIFAR-100 | - | $lr$: 0.1, $p$: 2.0, $\Omega$: 0.5, $\eta$: 0.5, $lr_{adam}$: 0.001 |
| | | Tiny-ImageNet | - | $lr$: 0.03, $p$: 2.0, $\Omega$: 0.5, $\eta$: 0.5, $lr_{adam}$: 0.001 |
| | | ImageNet-1000 | - | $lr$: 0.03, $p$: 2.0, $\Omega$: 0.5, $\eta$: 0.5, $lr_{adam}$: 0.001 |
| LwF | CIL/TIL | CIFAR-100 | - | $lr$: 0.03, $\alpha$: 0.5, $T$: 2.0, $wd$: 5e-5 |
| | | Tiny-ImageNet | - | $lr$: 0.03, $\alpha$: 0.5, $T$: 2.0, $wd$: 5e-5 |
| | | ImageNet-1000 | - | $lr$: 0.03, $\alpha$: 0.5, $T$: 2.0, $wd$: 5e-5 |
| EWC | CIL/TIL | CIFAR-100 | - | $lr$: 0.03, $\lambda$: 10, $\gamma$: 1.0 |
| | | Tiny-ImageNet | - | $lr$: 0.03, $\lambda$: 10, $\gamma$: 1.0 |
| | | ImageNet-1000 | - | $lr$: 0.03, $\lambda$: 10, $\gamma$: 1.0 |
| SI | CIL/TIL | CIFAR-100 | - | $lr$: 0.03, $c$: 0.5, $\xi$: 1.0 |
| | | Tiny-ImageNet | - | $lr$: 0.03, $c$: 0.5, $\xi$: 1.0 |
| | | ImageNet-1000 | - | $lr$: 0.03, $c$: 0.5, $\xi$: 1.0 |
| PNN | CIL/TIL | CIFAR-100 | - | $lr$: 0.03 |
| | | Tiny-ImageNet | - | $lr$: 0.03 |
| | | ImageNet-1000 | - | $lr$: 0.03 |
| GEM | CIL | CIFAR-100 | [1k, 2k] | $lr$: 0.03, $\gamma$: 0.5 |
| | | Tiny-ImageNet | [1k, 2k] | $lr$: 0.01, $\gamma$: 0.5 |
| | | ImageNet-1000 | [10k, 20k] | $lr$: 0.01, $\gamma$: 0.5 |
| | TIL | CIFAR-100 | [200, 500, 5120] | $lr$: 0.03, $\gamma$: 0.5 |
| | | Tiny-ImageNet | [200, 500, 5120] | $lr$: 0.03, $\gamma$: 0.5 |
| | | ImageNet-1000 | [200, 500, 5120] | $lr$: 0.03, $\gamma$: 0.5 |
| BiC | CIL | CIFAR-100 | [1k, 2k] | $lr$: 0.03 |
| | | Tiny-ImageNet | [1k, 2k] | $lr$: 0.03 |
| | | ImageNet-1000 | [10k, 20k] | $lr$: 0.2 |
| | TIL | CIFAR-100 | [200, 500, 5120] | $lr$: 0.03 |
| | | Tiny-ImageNet | [200, 500, 5120] | $lr$: 0.03 |
| | | ImageNet-1000 | [200, 500, 5120] | $lr$: 0.1 |
| PODNet | CIL | CIFAR-100 | [1k, 2k] | $lr$: 0.03, $\lambda_{pod}$: 1.0 |
| | | Tiny-ImageNet | [1k, 2k] | $lr$: 0.03, $\lambda_{pod}$: 1.0 |
| | | ImageNet-1000 | [10k, 20k] | $lr$: 0.03, $\lambda_{pod}$: 1.0 |
| | TIL | CIFAR-100 | [200, 500, 5120] | $lr$: 0.03, $\lambda_{pod}$: 1.0 |
| | | Tiny-ImageNet | [200, 500, 5120] | $lr$: 0.03, $\lambda_{pod}$: 1.0 |
| | | ImageNet-1000 | [200, 500, 5120] | $lr$: 0.03, $\lambda_{pod}$: 1.0 |
| DyTox | CIL | CIFAR-100 | [1k, 2k] | $lr$: 0.03 |
| | | Tiny-ImageNet | [1k, 2k] | $lr$: 0.1 |
| | | ImageNet-1000 | [10k, 20k] | $lr$: 0.1 |
| | TIL | CIFAR-100 | [200, 500, 5120] | $lr$: 0.03 |
| | | Tiny-ImageNet | [200, 500, 5120] | $lr$: 0.1 |
| | | ImageNet-1000 | [200, 500, 5120] | $lr$: 0.2 |
| Coil | CIL | CIFAR-100 | [1k, 2k] | $lr$: 0.03, $\lambda_{coil}$: 0.5 |
| | | Tiny-ImageNet | [1k, 2k] | $lr$: 0.03, $\lambda_{coil}$: 0.5 |
| | | ImageNet-1000 | [10k, 20k] | $lr$: 0.03, $\lambda_{coil}$: 0.5 |
| | TIL | CIFAR-100 | [200, 500, 5120] | $lr$: 0.03, $\lambda_{coil}$: 0.5 |
| | | Tiny-ImageNet | [200, 500, 5120] | $lr$: 0.03, $\lambda_{coil}$: 0.5 |
| | | ImageNet-1000 | [200, 500, 5120] | $lr$: 0.03, $\lambda_{coil}$: 0.5 |
| iCaRL | CIL | CIFAR-100 | [1k, 2k] | $lr$: 0.03, $wd$: 5e-5 |
| | | Tiny-ImageNet | [1k, 2k] | $lr$: 0.03, $wd$: 5e-5 |
| | | ImageNet-1000 | [10k, 20k] | $lr$: 0.03, $wd$: 5e-5 |
| | TIL | CIFAR-100 | [200, 500, 5120] | $lr$: 0.03, $wd$: 5e-5 |
| | | Tiny-ImageNet | [200, 500, 5120] | $lr$: 0.03, $wd$: 5e-5 |
| | | ImageNet-1000 | [200, 500, 5120] | $lr$: 0.03, $wd$: 5e-5 |

Table 13: Selected best hyperparameters for all methods across datasets- Part II.

| Method | Scenario | Dataset | Buffer | Selected Hyperparameters |
|---|---|---|---|---|
| DER | CIL | CIFAR-100 | [1k, 2k] | $lr$: 0.03, $\alpha$: 0.3 |
| | | Tiny-ImageNet | [1k, 2k] | $lr$: 0.03, $\alpha$: 0.3 |
| | | ImageNet-1000 | [10k, 20k] | $lr$: 0.03, $\alpha$: 0.3 |
| | TIL | CIFAR-100 | [200, 500, 5120] | $lr$: 0.03, $\alpha$: 0.5 |
| | | Tiny-ImageNet | [200, 500, 5120] | $lr$: 0.03, $\alpha$: 0.5 |
| | | ImageNet-1000 | [200, 500, 5120] | $lr$: 0.03, $\alpha$: 0.5 |
| DER++ | CIL | CIFAR-100 | [1k, 2k] | $lr$: 0.03, $\alpha$: 0.2, $\beta$: 0.5 |
| | | Tiny-ImageNet | [1k, 2k] | $lr$: 0.03, $\alpha$: 0.2, $\beta$: 1.0 |
| | | ImageNet-1000 | [10k, 20k] | $lr$: 0.03, $\alpha$: 0.2, $\beta$: 1.0 |
| | TIL | CIFAR-100 | [200, 500, 5120] | $lr$: 0.03, $\alpha$: 0.1, $\beta$: 0.5 |
| | | Tiny-ImageNet | [200, 500, 5120] | $lr$: 0.03, $\alpha$: 0.2, $\beta$: 0.5 |
| | | ImageNet-1000 | [200, 500, 5120] | $lr$: 0.03, $\alpha$: 0.2, $\beta$: 1.0 |
| MEMO | CIL | CIFAR-100 | [1k, 2k] | $lr$: 0.03 |
| | | Tiny-ImageNet | [1k, 2k] | $lr$: 0.1 |
| | | ImageNet-1000 | [10k, 20k] | $lr$: 0.1 |
| | TIL | CIFAR-100 | [200, 500, 5120] | $lr$: 0.03 |
| | | Tiny-ImageNet | [200, 500, 5120] | $lr$: 0.1 |
| | | ImageNet-1000 | [200, 500, 5120] | $lr$: 0.1 |

