# OpenReview forum: "Less Forgetting Learning: Memory-free Continual Learning Classification"
_ICLR.cc/2026/Conference — ICLR 2026 Conference Withdrawn Submission_

### Official Review · Reviewer_M85P · 2025-10-15

**Soundness:** 2
**Presentation:** 2
**Contribution:** 2
**Rating:** 2
**Confidence:** 5

**Summary:**

This paper proposes a continual learning framework, LFL, and its enhanced version, LFL+, to mitigate catastrophic forgetting. The core contribution lies in its stepwise freezing and training strategy. The model is decomposed into a shared feature extractor and task-specific heads. When learning a new task, LFL controls the updates of different components through several training phases. First, the shared parameters and old task head are frozen while only the new task head is trained to adapt to the new task; next, the new task head is frozen, and the shared parameters and old task head are aligned using soft targets; finally, a joint fine-tuning phase is performed to consolidate all learned knowledge. LFL+ further integrates an autoencoder to constrain and preserve essential feature representations.

**Strengths:**

1. This paper focuses on an important and widely discussed problem in the field of continual learning and proposes a memory-free approach.
2. The paper is generally clearly written, making it easy to follow the main ideas.

**Weaknesses:**

1. The Introduction and Related Work sections provide a broad overview of Continual Learning. While this is useful for context, they fail to frame and motivate the specific contribution of this work. It isn't easy to discern the key differences between this paper and other memory-free continual learning methods.
2. The proposed 4- or 5-phase training paradigm lacks details on how each phase is conducted. In the experiments, only a generic grid search is reported. Given that even standard end-to-end training methods can exhibit substantial variability, the absence of practical guidance on hyperparameter selection in this multi-phase framework limits its generalizability and practical applicability to other domains.
3. LFL may heavily rely on a similar dataset or task settings. One of the core ideas of LFL is to use the output produced by the old model on new data as soft targets. However, if the data distributions of the old and new tasks differ significantly, the targets generated by the old model on the new data may be meaningless or even noisy. In such cases, forcing the new model to fit these noisy signals may actually degrade performance.
4. The shared encoder is described as a “teacher,” but in practice, it serves as a shared representation for both old and new tasks. During the third training phase, if the shared encoder is updated and its internal representations drift, this may lead to irreversible degradation in performance on the old tasks.
5. Although the autoencoder introduced in LFL+ is intended to preserve feature representations, it also requires training a separate autoencoder for each task. Since autoencoder training itself is a relatively complex and sensitive optimization process, incorporating it into an already complicated five-phase training pipeline makes the overall approach difficult to justify in practice.
6. Many recent studies have shown that storing a small number of representative samples (i.e., a small fixed memory buffer) is generally acceptable and can mitigate catastrophic forgetting in a simple and highly effective manner, with much lower implementation difficulty and computational cost. LFL incurs significant engineering complexity to address a problem that could likely be solved more efficiently with a small memory buffer and straightforward replay.

**Questions:**

Please refer to the Weakness section for detailed comments.

---

> ### Author Response · Authors · 2025-11-16
> **Response to Reviewer M85P**
>
> Thank you for your feedback.
>
> Response to Weakness 1
>
> Lines 63–68 describe the gap between our approach and prior continual learning methods. Lines 79–86 explain our motivation for adopting a stepwise freezing strategy, and lines 87–93 summarize our main contributions. The central distinction we emphasize is that our method relies on a structured stepwise freezing procedure rather than architectural changes or memory buffers. We also provide Table 2, where the difference due to the page limit is summarized shortly with other methods.
>
> Response to Weakness 2
>
> We use a grid search for hyperparameter selection, as stated in the paper. The full implementation of this procedure is provided both in the code repository and in the Docker image. Because of page constraints, we do not include an extended description of grid search. We respectfully note that the concern about “absence of practical guidance” is not fully applicable here, since the selection method is clearly specified and the full configuration is available in the implementation.
>
> Response to Weakness 3
>
> We agree that if the distributions of old and new tasks differ substantially, the soft targets produced by the old model may become less informative. This limitation affects all knowledge-distillation-based methods, including LwF, iCaRL, BiC, and DER, not only LFL. In cases of severe distribution shift, memory-based or architecture-based approaches are indeed stronger options. Our method shares this inherent limitation with other memory-free strategies.
>
> Response to Weakness 4
>
> Our work does not use or refer to a “shared encoder” in the sense implied. In LFL+, the autoencoder is used to preserve feature representations for each task. Each autoencoder (W_{\text{enc}}) is task-specific, so updates from new tasks do not modify previously trained encoders. Representation drift is a challenge in most continual learning methods, but in our case the reconstruction loss in Eq. 12 encourages the model to approximate the old inputs rather than the drift introduced by later tasks.
>
> Response to Weakness 5
>
> We agree that training a separate autoencoder for each task adds complexity and may not be feasible in all settings. This is why step 5 is presented as LFL+ and not part of the core LFL pipeline. In scenarios where autoencoder training is impractical, we recommend using LFL alone. We note that prior work has explored autoencoder-based continual learning directly [1], so the idea is consistent with established approaches.
>
> Response to Weakness 6
>
> We respectfully disagree with the characterization of this point as a weakness. The goal of this work is to explore memory-free continual learning, which is explicitly motivated in lines 64–66. In lines 64-66, we argue that relying on memory violates the true definition of CL. In many practical environments, storing past data is not allowed due to privacy regulations, ownership restrictions, or deployment constraints. For these scenarios, memory-free methods remain essential. LFL introduces no memory buffer and requires less training time than many architecture-based or memory-based approaches. What complexity do we introduce? Our training time is less than the architecture-based and memory-based; our memory need is zero, our model buffer is 2.53 of SOTA. If this is our work weakness, we can raise the same questions about why, just saving some memory and trying to resolve catastrophic forgetting, why not save all data from previous tasks and completely remove catastrophic forgetting?
>
>
> [1] Rannen et al. Encoder-based Lifelong Learning. ICCV, 2017.

---

> > ### Author Response · Authors · 2025-11-23
> > **Modifications Response to Reviewer M85P**
> >
> > We have updated the paper based on the changes requested by you and the other reviewers. Please review the revised version that we have uploaded for the rebuttal phase.
> >
> > Modification regarding Weakness 1:
> > We added Subsection 3.1 to clarify the motivation and contribution of each step in our method.
> >
> > Modification regarding Weakness 2:
> > We added Section I in the Appendix to present the hyperparameter search space and the selection process. We also revised lines 408–425 to include the necessary details about the validation set and the training procedure.
> >
> > Modification regarding Weaknesses 3, 4, 5, and 6:
> > These points were addressed in our first-round response. Please refer to lines 411–414 and our earlier explanation for Weakness 5. For Weakness 6, we would like to emphasize again that there exist many memory-free methods, and it is not appropriate to devalue such approaches simply because they avoid using memory buffers. This constraint is essential in many real-world domains where storing user data is not acceptable.
> >
> > If our revisions meet your expectations, we kindly ask you to reconsider your score.

---

### Official Review · Reviewer_9HJx · 2025-10-27

**Soundness:** 3
**Presentation:** 2
**Contribution:** 2
**Rating:** 4
**Confidence:** 3

**Summary:**

Continual learning (CL) deals with the incremental training of neural networks from a data stream. Owing to the nature of gradient-based optimization, catastrophic forgetting (CF) occurs: the performance on old tasks drops as newer tasks are learned. To mitigate forgetting, authors propose knowledge distillation by using the output of a network from the end of the previous task to regularize the updates on the current task. They pair this with a slight parameter expansion method of adding task-specific heads. They demonstrate the strong performance from their method on common CL datasets and in two settings (class- and task-incremental learning).

**Strengths:**

- studied both CIL and TIL
- extensive set of experiments

**Weaknesses:**

- limited novelty: method is a combination of existing strategies (mainly, knowledge distillation with task-specific heads)
- Paper focuses on memory-free CL, but benchmarks against a large set of memory-based methods (which are outperformed mostly tho) -- authors are encouraged to add 2 to 3 recent memory-free methods into their comparison
- Plasticity stability ratio has priorly been introduced [1] (please comment on differences)
- Presentation and description of the algorithm should be updated for improved readability; there are a lot of subscripts, superscripts, underline, etc.
- Its unclear how the method works in CIL: There are no task-IDS available to select the head, because only one head is expanded in CIL. So, there is only one head available in CIL, which makes doing steps 2, 3, 4 impossible?
- ablation study on the individual components/ steps are missing

**Questions:**

- How does the method work in the CIL setup?
- line 301: what is the cross-validation set? Wouldnt it require a memory buffer?

**Suggestions:**
- Table 2 does not add, maybe move to appendix
- use correct \cite commands (currently misses brackets around references in the text, eg line 37)
- improve figure 1: integrate legend into figure
- I think that the authors do not make themselves a favour by benchmarking agains memory-based methods. I'd suggest sticking to the memory-free regime for a fairer comparison, and benchmark agains memory-free methods only (such as already done with EWC)

---

> ### Author Response · Authors · 2025-11-15
> **Response to Reviewer 9HJx**
>
> Thank you for your detailed review and constructive feedback.
>
> Response to Weakness 1:
> Our main contribution is the "stepwise freezing" strategy. This method is a strategy for using the potential of each network instead of introducing a new architecture or buffer.  Each 3-component may be frozen or trained, producing eight configurations in total. We select the sequence that minimizes interference across stages so that each step preserves what earlier steps have learned. The approach relies on overparameterization, motivated by findings from the Lottery Ticket Hypothesis [1], and uses the available parameters more intentionally than prior work.
>
> Response to Weakness 2:
> The contribution is not only in the "stepwise freezing" terminology but in how each step forces the network to use the available parameters as effectively as possible. Most memory-free methods do not attempt to activate or constrain every parameter in this way. Our approach strongly depends on the capacity of the original network. If we apply weight or neuron pruning after each step, the performance drops noticeably, which suggests that the method uses most of the available capacity. Many existing methods can be compressed to ten to twenty percent of their size without large losses, which indicates a different use of network capacity. This could also be in supplementary material as a pruning study for CL study.
>
> We already include several memory-free baselines and plan to add PEC [2] and WSN [3] for the camera-ready version to strengthen the comparison.
>
> Response to Weakness 3:
> Could you clarify the specific paper you are referring to?
>
> Response to Weakness 4:
> While there is always room for improvement. The nature of the method is complex to present with mathematical notation, but easy to explain in Figure 1. We tried to be clear without any notation interfering with each other. However, we will revise the notation, simplify subscripts and superscripts where possible, and polish the algorithm description in the camera-ready version. We also added Table 1 to make it more readable.
>
> Response to Weakness 5:
> This experiment is included in Section F of the supplementary material.
>
> Response to Questions
>
> Q1.
> The single head is a unified structure that holds both the old and new task submodules. The new task loss updates the new submodule, while the distillation loss anchors the old submodule. This allows steps 2, 3, and 4 to function even though the output layer is presented as one head during inference.  In CIL, the single head is not static. It expands and internally contains the roles of both the old and new heads. During training, the neurons associated with previous tasks is guided by the distillation loss, while the newly added neurons are driven by the new task loss. Task IDs are needed only during training to compute these losses. They are not required at inference time.
>
> Q2.
> This set is only used during training for model selection. It is not stored for future tasks, so it does not function as a memory buffer. The same assumption appears in many continual learning methods that rely on a validation split within the current task.
>
>
>
> Response to Suggestions
>
> We agree that Table 2 and the figure improvements can be addressed in the camera-ready version. Regarding citations, we follow the ICLR template, which does not place brackets around inline references. We also agree that memory-based methods can be moved to the appendix to emphasize the memory-free comparison in the main text.
>
> References:
>
> [1] Frankle and Carbin. The Lottery Ticket Hypothesis. 2019.
>
> [2] Zajac. Prediction Error Based Classification for Class Incremental Learning. ICLR 2024.
>
> [3] Kang. Forget Free Continual Learning with Winning Subnetworks. ICML 2022.

---

> > ### Comment · Reviewer_9HJx · 2025-11-18
> >
> > I thank the authors for the detailed responses to my questions. I encourage them to include the answers to my questions (single head setting, usage of a validation set) into a revised submission for clarification.
> >
> > Regarding the missing citation (Response to weakness 3): Elsayed, M., & Mahmood, A. R. Addressing Loss of Plasticity and Catastrophic Forgetting in Continual Learning. In The Twelfth International Conference on Learning Representations, 2024
> >
> > This paper introduces a stability-plasticity tradeoff, please comment on the differences.
> >
> > At the current moment, I'll wait for a revised version until I might reconsider my score. Specifically, I am looking for a revised section 3 and improved Figure 1.

---

> ### Author Response · Authors · 2025-11-22
> **Second Round Response to Reviewer 9HJx**
>
> Thank you for the constructive discussion during the rebuttal phase. We have updated the paper based on the changes you and the other reviewers requested. Please check the revised version that we have uploaded for the rebuttal phase.
>
> Regarding Weaknesses 1 and 2:
> We added Subsection 3.1 to provide the motivation behind each step of our method.
>
> Regarding Weakness 4:
> We revised and simplified Figure 1, including its notation, color coding, and caption, to improve clarity. We also converted several equations into descriptive text and added explanations to make the section more readable. The pseudocode in Section 3 now complements the explanations and avoids repeating equations.
>
> Regarding Weakness 5:
> We addressed your question in ines 356–367.
>
> Regarding Weakness 6:
> As mentioned in the first-round rebuttal, Section F already provides the requested analysis. In addition, we added Subsection F.1 with an ablation study on parameter usage in our method.
>
> Following your suggestions, we also moved Table 2 to the appendix and changed all citation commands to \citep.
> Furthermore, for the camera-ready version, we plan to include WSN and PEC as memory-forgetting methods in our benchmark. Their implementations are already in progress.
>
>
> Regarding Weakness 3:
> 1. Our metric is designed for task-based continual learning with separate, well-defined tasks and offline evaluation, while the referenced metric is created for online continual learning.
>
> 2. Our metric models plasticity and stability together as a ratio, whereas the referenced metric measures only plasticity based on immediate loss change for the current sample.
>
> 3. We define plasticity in terms of accuracy improvement, while the referenced work defines it as the ability to reduce the loss on a specific data point.
>
> 4. Our denominator uses the absolute value of the backward transfer metric (BWT) to measure how much the model forgets after learning future tasks. The forgetting measure in the referenced work is conceptually closer to the intransigence metric (I) described in Appendix C of our paper. Originally, it was defined in [1].
>
> Overall, our metric captures the trade-off between plasticity and stability in a single expression aimed at offline, accuracy-based evaluation. In contrast, the referenced metric is tailored for online continual learning and is computed on a window of samples.
>
> If our revisions meet your expectations, we kindly ask you to reconsider your score. Thank you again for your time and constructive feedback.
>
> [1] A. Chaudhry et al., Riemannian Walk for Incremental Learning: Understanding Forgetting and Intransigence, ECCV 2018.

---

> > ### Comment · Reviewer_9HJx · 2025-11-24
> >
> > I thank the authors for preparing an updated manuscript. The newly added Sec. 3.1 is helpful, especially the T(rain) and F(reeze) abbreviations. This should also be integrated into Figure 1, maybe as a little text next to the current numbering (1), (2), etc. I also thank for the comparison to the related work I asked for; it makes the differences clearer.
> > I am raising my score.

---

> > > ### Author Response · Authors · 2025-11-24
> > >
> > > We sincerely appreciate your constructive feedback and for raising the score.
> > > We will also incorporate your suggestion in Figure 1.

---

### Official Review · Reviewer_q7UX · 2025-10-29

**Soundness:** 3
**Presentation:** 2
**Contribution:** 3
**Rating:** 4
**Confidence:** 5

**Summary:**

The paper proposes Less Forgetting Learning (LFL), and its enhanced variant LFL+, which are memory-free Continual Learning (CL) methods designed to mitigate Catastrophic Forgetting without relying on an explicit memory buffer. LFL involves a multi-step freezing and fine-tuning training protocol incorporating knowledge distillation using previous target signals. The enhanced variant, LFL+, incorporates an auto-encoder to preserve task-relevant features, along with a bias correction mechanism to mitigate classification head and bias toward new classes. Experiments on CIFAR-100, Tiny-ImageNet and ImageNet-1000 under Class-Incremental Learning (CIL) and Task-Incremental Learning (TIL) show competitive performance compared to recent buffer-based methods,  with improved scalability and shorter training time.

**Strengths:**

- The proposed method addresses a well-know problem in CL, namely the need for effective methods that do not rely on the use of a buffer to store samples from previous tasks.

- The method contributes a clear and simple stepwise freezing and fine-tuning procedure addressing the stability-plasticity dilemma in CL

- In the LFL+ variant, the use of the autoencoder to preserve the features of the previous task is reasonable, because it overcomes the problem of forgetting on the autoencoder itself, as it is not continual trained on the sequence of tasks, but re-trained task by task only on recent features.

- The bias correction mechanism addresses class imbalance in a principled manner to enhance performance in class incremental learning.

- Comprehensive empirical evaluation compares LFL and LFL+ against a range of strong baselines and across multiple standard benchmarks.

**Weaknesses:**

- While the overall paper is generally easy to follow, the Method section (Section 3) is often unclear and ambiguous. For instance, the statement "the new task head is kept frozen, while the feature extractor and the previously learned task head are updated during training after random initialization" (line 227) creates confusion between random initialization and parameter fine-tuning. Ambiguity is also introduced through figures and pseudo-code notation, which do not always clarify the flow between "randomly initialized/updated/fine-tuned parameters", and frozen parameters. Clearer language and explicit step-by-step descriptions would help readers precisely reproduce the pipeline.

- The “memory-free” claim is repeated frequently and highlighted as a key strength versus competitors. However, while LFL/LFL+ do not store real data exemplars from previous tasks, they do require a temporary buffer to store logits from previous models for knowledge distillation. This memory requirement, although far less than storing raw images, is intrinsic to the approach and not explicitly stated.

- The discussion of cross-validation details (used for bias correction in LFL+) is not adequately described, limiting reproducibility.

- Unclear training procedure: the paper does not explicitly indicate a fixed number of epochs per task in the training procedure. The description refers generically to “training until convergence” (line 217) for each step or task, without providing precise details on the epoch count per task. Therefore, the paper seems to suggest that the number of epochs per task is treated as a hyperparameter defined empirically or by convergence conditions, not strictly defined in the experimental protocol.

- In the experimental setup, using different buffer sizes for Class-IL and Task-IL methods may introduce a form of bias in the evaluation, making results hard to compare “on equal ground". As a buffer-free method, LFL/LFL+ results are relevant and competitive, but choosing different amount of memory for buffer-based methods in Task-IL could artificially disadvantage them. In addition, comparison between Task-IL results in this paper and those in the DER++ paper reveals differences for several shared methods, raising concerns about reproducibility.

- Section 4.3 (Training Time Comparison) raises further concerns:

 i) The reported training time difference between CL-IL and Task-IL is surprisingly large. For buffer-based methods like DER++, the training pipeline is virtually identical for CL-IL and Task-IL, except for access to task-ID at inference, which should not substantially affect runtime. Such large training time differences may only be justified if different numbers of training epochs were used, which again links back to ambiguity about the training protocol.

ii) The training time difference is attributed to buffer management and dynamic model complexity. While generally reasonable, buffer size mainly impacts specific methods. For example, DER++ retrieves the same number of buffer samples per stream batch regardless of buffer size.

iii) It is unclear how LwF, which essentially corresponds to step 1 of the proposed method, can have longer training time than LFL on CIFAR-100.

**Questions:**

Please address and resolve the concerns raised in the Weaknesses section.

---

> ### Author Response · Authors · 2025-11-16
> **Response to Reviewer q7UX**
>
> Thank you for your detailed feedback and review.
>
> Response to Weakness 1
>
> While there is always room for improvement, we included Table 1 as a reference for each notation. In addition, we provided descriptive captions for the five steps, where we clearly show which parts are trained (green) and which parts are frozen (grey). We also state that the output of each step serves as the input to the next step. Throughout Section 3, we tried to isolate each notation and parameter, and equations such as Eq. 3, 4, 6, and 10 show the components trained at each stage.
>
> We decompose the network into three components: (1) a shared backbone, (2) the old task head, and (3) the new task head. Each component can be frozen or trained, creating 8 possible configurations. We chose the steps that interfere with previous updates as little as possible so that each stage preserves what earlier stages learned. Overall, the pipeline is conceptually simple, though the math introduces some complexity. We agree that clarity can be improved, but we ask readers to consider Table 1 and Figure 1 when following the method. Also, in the code, each step is fully separated.
>
> Response to Weakness 2
>
> In the abstract, we highlighted the distinction between the model buffer and the memory buffer (lines 17 and 25).
> In lines 72–73 we wrote: “Memory-free means they do not rely on any external memory buffer (i.e., data from previous tasks); nevertheless, they employ a model buffer.”
> Table 4 also distinguishes between the model and the memory buffer. We never claimed the method is model-free; we only stated it is memory-free, and by memory we refer strictly to the memory buffer.
>
> Response to Weakness 3
>
> We argue that this point is related to the code. In the code, the cross-validation set is explicitly defined. We also included relevant details in Sections H and D to help with understanding the training phases. We can add an explanation of the cross-validation split in the appendix for the camera-ready version. It is simply 10 percent of the training set, which can be added to the manuscript.
>
> Response to Weakness 4
>
> Similar to Weakness 3, we see this as a code-related detail rather than part of the core research description. All such details are available in the public code. As noted, we suggest training until convergence. Explaining convergence in detail would be repetitive and unnecessary, but we can add this to the appendix if needed.
>
> Please note that Weaknesses 3 and 4 both ask for implementation-level details to appear in the manuscript. We agree that more explanation reduces ambiguity, but a research paper is anchored by the implementation, and some details can remain in the code. We can add a hyperparameter selection section and the statement that the validation set is 10 percent of the training set to the appendix. Thank you for noting the Weaknesses 3 and 4 as simply applied right away for removing ambiguity.
>
> Response to Weakness 5
>
> In lines 360–362, we explicitly state:
> “For memory-based methods in the CIL scenario, we set the number of exemplars to 1,000 and 2,000 for CIFAR-100 and Tiny-ImageNet, and also 10,000 and 20,000 for ImageNet-1000 [1]. In the TIL scenario, we set 200, 500, and 5120 exemplars for all datasets [2].”
>
> Reference [2] in the review corresponds to the DER++ paper.
> We did not disadvantage any method; we followed the exact training procedure established in DER++. DER++ used ResNet-18 for TinyImageNet. We used He initialization, while DER++ used random initialization and averaged over 10 runs. This explains the slight difference for TinyImageNet, but the code is public, so reproducibility is ensured.
>
> Response to Weakness 6
>
> The difference in training time is not due to the number of classes but due to the complexity of the optimization objective and the loss function for Class-IL. This trend is also shown in [4], figure 30, and subfigure (f) in figure 2 of the DER++ paper you mentioned, which reports similar patterns on different datasets.
>
> Overall, LwF has a shorter training time than LFL except on CIFAR-100. LwF includes an extra regularization term (weight decay) and uses Xavier initialization. LwF is also not analogous to step 1 of our method; it trains both heads and the backbone simultaneously, not separately as we do. LwF also includes a warmup step (similar to our step 2), but they did not provide any ablation study on how that warmup affects training time or accuracy. Our ablation (Table 9) shows this effect is significant, not negligible. Also, differences between He and Xavier initialization have more impact on smaller datasets.
>
> References
>
> [1] Da-Wei Zhou, Class-Incremental Learning: A Survey, TPAMI, 2024
>
> [2] Pietro Buzzega, Dark Experience for General Continual Learning: A Strong, Simple Baseline, NeurIPS, 2020

---

> ### Author Response · Authors · 2025-11-22
> **Modifications Response to Reviewer q7UX**
>
> We have updated the paper based on the changes requested by you and the other reviewers. Please check the revised version that we have uploaded for the rebuttal phase.
>
> Modification regarding Weakness 1:
> We revised Section 3 entirely and added Subsection 3.1. We also updated Figure 1 and its notation. These changes were made to improve clarity and make the proposed method more readable and compact.
>
> Modification regarding Weakness 2:
> Please refer to our response from the first round, as well as lines 72–73 in the revised manuscript.
>
> Modification regarding Weaknesses 3 and 4:
> Lines 408–425 have been revised to include the necessary information about the validation set and the training procedure. We also added Section I in the Appendix, which describes our hyperparameter search space and selection process.
>
> Modification regarding Weaknesses 5 and 6:
> These points were fully addressed in our first-round response. Please refer to that explanation.
>
> Modification regarding your question:
> Please refer to Subsection F.1 for additional ablation studies comparing LwF and our method. Further details are also available in our first-round response.
>
> If our revisions meet your expectations, we kindly ask you to reconsider your score.

---

### Official Review · Reviewer_N6JE · 2025-10-30

**Soundness:** 2
**Presentation:** 2
**Contribution:** 2
**Rating:** 2
**Confidence:** 4

**Summary:**

This work develops a Less Forgetting Learning (LFL) paradigm to balance plasticity and stability in continual learning. Specifically, LFL is performed with four steps, each of which is based on the previsous output. Besides. LFL+ integrates an additional Auto-Encoder for feature retention further. Extensive results on CIFAR and ImageNet datasets show the effectiveness of the proposed LFL.

**Strengths:**

This work aims to devise a exemplar- free continual learning method, which does not rely on memory buffer.

The implementation details on the four steps in LFL are provided, and it might be easy to reproduce the method properly.

The experimental evaluations are comprehensive and thorough. Besides, training cost is compared as well.

**Weaknesses:**

The techinal contributions are not significant, as the proposed LFL is mainly composed of several freezing and fine-tuning steps. It is hard to identify the new insights on continual learning.

After reading the four steps in LFL, I am confused by the relations between them. Specifically, it is unclear why step 2 and step 3 are separated from each other. In general, it is reasonable to fine-tune the feature extractor and train the new classifier at the same time, given the knowledge distillation regularization. It is needed to clarify the strength of learning them in two steps. Moreover, what is the motivation behind step 4?

The objective loss function for step 5 (as in Eq.17) is too complex, which may make it hard to adapt to various scenarios.

**Questions:**

Some experimental results are not convining. For example, in Table 9, the ablation model with step 2 even achieves 38.92 accuracy score, which outperforms SI and EWC, and competes with LwF, as shown in Table 3. It is hard to understand about why step 2 can bring such remarkable improvements, as it is simply freezing the backbone and training new classifier from scratch.

---

> ### Author Response · Authors · 2025-11-15
> **Response to Reviewer N6JE**
>
> Thank you for your review and for the detailed feedback.
>
> Response to Weaknesses 1 and 2:
>
> Our main motivation comes from the role of overparameterization in neural networks. The Lottery Ticket Hypothesis [1] shows that large networks can often be compressed to ten to twenty percent of their size without losing performance. We examined whether this property can support continual learning by improving stability during training instead of introducing new architectures or relying on external memory.
>
> We decompose the network into three components: (1) a shared backbone, (2) the old task head, and (3) the new task head. Each component can be frozen or trained, creating 8 possible configurations. We selected the steps that interfere with previous updates as little as possible so that each stage preserves what earlier stages learned.
>
> Steps 2 and 3 are separated intentionally. Joint training of the feature extractor with both heads can cause updates from the new head to influence the old head’s representation. We isolate these steps to avoid this interference. In step 4, we then allow both heads to update the backbone together. The goal is to let the shared representation settle with guidance from both tasks. The two soft targets with different initializations help anchor performance to the old task while still allowing adaptation to the new one. This design aligns with observations on bias toward new classes in large incremental classification [2].
>
> Response to Weaknesses 3:
>
> We agree that the loss in step 5 is complex. This is a limitation of our extended method. For this reason, we present step 5 as LFL+ rather than part of the core LFL pipeline. In settings where training an autoencoder is difficult or costly, we recommend using LFL without step 5.
>
> Response to the Question:
>
> Thank you for examining the supplementary results closely. The strong performance of step 2 in Table 9 aligns with observations in the LwF paper, which also includes a warmup step (Step 2) but does not provide an ablation study for it. Our experiments show that this warmup contributes significantly. Why?. Freezing the backbone and training a new classifier forces the network to use every available neuron and weight in the representation. Evidence from pruning literature supports this behavior. "During retraining, it is better to retain the weights from the initial training phase for the connections that survived pruning than it is to re-initialize the pruned layers...gradient descent is able to find a good solution when the network is initially trained, but not after re-initializing some layers and retraining them." [3].
>
>  Step 2 benefits from this effect, acting as a focused finetuning stage that strengthens the existing representation rather than relying on randomly initialized parameters as in LwF. Overall, our approach emphasizes retraining and stabilizing each component as much as possible. This helps explain the performance gain observed for step 2. This question will raise an idea about doing pruning as another ablation study for each step for camera-ready.
>
>
> [1] J. Frankle and M. Carbin. The Lottery Ticket Hypothesis. ICLR 2018.
>
> [2] Yue Wu, Large Scale Incremental Learning. CVPR 2019.
>
> [3] Song Han. Learning both weights and connections for efficient neural network. NeurIPS, 2015.

---

> ### Author Response · Authors · 2025-11-22
> **Modifications Response to Reviewer N6JE**
>
> We have updated the paper based on the changes requested by you and the other reviewers. Please check the revised version that we have uploaded for the rebuttal phase.
>
> Modification regarding Weaknesses 1 and 2: We added Subsection 3.1 to explain the motivation and overall contribution of each step. We also revised Section 3 in its entirety to improve readability.
>
> Modification regarding Weakness 2 (additional clarification): We added Subsection F.1 in the Appendix to analyze the contribution of our method by comparing it with LwF. This highlights the potential of our approach in parameter utilization and demonstrates the effectiveness of the new stepwise freezing strategy.
>
> Modification regarding Weakness 3: In line 413, we added a discussion about the Autoencoder size and referred back to our explanation provided in the first-round response.
>
> If our revisions meet your expectations, we kindly ask you to reconsider your score.

---

### Comment · Area_Chair_4vyp · 2025-11-22

Dear reviewers,
Please check the authors’ responses. As there are differing opinions about the paper, it would be appreciated if you could evaluate—based on all comments—whether the authors have adequately addressed the main concerns.
Br,

---

### Note · Authors · 2026-01-29

I have read and agree with the venue's withdrawal policy on behalf of myself and my co-authors.

---

### Meta-Review · Area_Chair_5pF5 · 2025-12-29

**Summary:**

This submission proposes LFL, a memory-free continual learning training protocol that mitigates forgetting via a multi-phase, stepwise freezing/fine-tuning schedule with distillation, and optionally augments it with task-wise autoencoders and bias correction. Reviewers’ main concerns centered on limited novelty beyond combining known ingredients, substantial ambiguity in the training pipeline and notation (especially the separation/motivation of phases and the CIL “single-head” interpretation), and insufficiently specified experimental protocol details that affect reproducibility. There was also skepticism about practical value versus simpler replay baselines, and questions about robustness under distribution shift given reliance on old-model soft targets. Overall, the post-rebuttal picture is that the paper is likely technically sound and empirically competitive, but it initially suffered from clarity and protocol-specification gaps that drove divergence in reviewer confidence and perceived contribution.

**Reviewer Concerns:**

The rebuttal and revised manuscript appear to have meaningfully addressed the most actionable concerns about clarity and missing protocol details: the authors added a motivating Subsection 3.1, revised Section 3 and Figure 1, clarified the CIL single-head mechanism (internal submodules guided by task loss and distillation during training without task-ID at inference), and incorporated explicit information about the validation set usage, training procedure, and hyperparameter search space into the paper/appendix, which directly targets reproducibility objections. The discussion about the stability–plasticity metric and the distinction from the referenced ICLR 2024 work was also acknowledged as improved by the requesting reviewer, reducing the “missing related work / unclear difference” concern. However, some issues remain only partially resolved: novelty is still viewed as incremental because the core method is a careful orchestration of freezing and distillation rather than a fundamentally new principle, and the practicality critique (engineering complexity vs. small-buffer replay) persists as a positioning question rather than something the rebuttal can fully eliminate. Likewise, reliance on soft targets under severe distribution shift remains an inherent limitation shared with distillation-based memory-free methods, and while the authors correctly contextualize it, it is still an outstanding boundary condition for the approach.

**Reviewer Scores:**

Although Reviewer 9HJx indicated an intention to raise their score after seeing the revision, I would treat this as a limited adjustment rather than a decisive shift, because their original critique emphasized missing comparisons, unclear CIL mechanics, and presentation issues—points that can be improved without materially changing the underlying novelty assessment. Reviewer q7UX’s concerns around ambiguity and experimental protocol are partially addressed, but their deeper reproducibility and fairness reservations (e.g., consistency with prior reports and the interpretability of training-time claims) plausibly remain, so I would not expect a strong upward change in their score. Reviewer N6JE’s primary objections target the perceived lack of significant technical insight and the complexity/motivation of the staged design; these concerns are only modestly alleviated by added explanation and are unlikely to translate into a meaningful score increase. For Reviewer M85P, regardless of the authors’ allegation about review quality, the stated weaknesses include several broad practicality/positioning critiques that are not convincingly overturned by the rebuttal, so I would not anticipate any upward revision. Overall, I would expect at most minor softening from one or two reviewers rather than a substantial score raise, with the review set remaining anchored by novelty and practicality concerns.

---

### Decision · Program_Chairs · 2026-01-26

Reject